# GradientHide: Federated Learning with Two-Stage Local Update for Defending Against Gradient Inversion Attacks

## Abstract

Federated learning enables collaborative training of neural networks across distributed clients coordinated by a central server. In each communication round, clients receive the current global model parameters and upload gradient updates computed on their private local data. However, transmitting such updates poses significant privacy risks, as adversaries may exploit them to reconstruct sensitive training data via gradient inversion attacks. To address this challenge, we propose GradientHide, a novel defense framework that obfuscates private information contained in gradients. Specifically, we introduce an additional update step using public data before transmitting gradients to the server, thereby hiding privacy information embedded in gradients. To mitigate potential performance degradation from using public data, we leverage CLIP's zero-shot inference for semantic alignment, enabling effective use of public images without extra training. GradientHide is evaluated against representative gradient inversion attacks and compared with state-of-the-art defense approaches across three benchmark datasets, followed by a thorough analysis of its effectiveness. Our findings demonstrate that GradientHide offers substantial resistance to gradient inversion attacks, evidenced by lower PSNR scores and semantic distortion in reconstructions, while preserving competitive model performance.

## 1 Introduction

Federated learning (FL) (Kairouz et al., 2021; McMahan et al., 2017) is an increasingly prevalent decentralized training framework, minimizing direct data transmission and enhancing privacy protection. Specifically, clients train models locally on their private data, and then transmit model parameters or gradients to a central server for aggregation of these updates (*i.e.*, model parameters or gradients) to refine a global model.

Although FL prevents each client's data from being transmitted directly, recent studies (Geiping et al., 2020; Yin et al., 2021; Li et al., 2022; Jeon et al., 2021; Fang et al., 2023; Ye et al., 2024; Sami et al., 2025; Yu et al., 2025) have demonstrated that gradient inversion attacks (GIAs) can reconstruct training samples from shared gradient updates using optimization. Potential adversaries, such as honest-but-curious servers (Phong et al., 2017) or eavesdroppers (Xu et al., 2025), can intercept these updates to recover sensitive information. Previous studies, including Gaussian noise injection for differential privacy (Geyer et al., 2017; Yuan et al., 2023; Li et al., 2024), gradient clipping (Wei et al., 2021b), gradient sparsification (Aji & Heafield, 2017), Soteria (Sun et al., 2021), and CENSOR (Zhang et al., 2025), have been proposed to defend against GIAs.

Besides gradient inversion defenses (GIDs), several approaches aim to directly obfuscate private data. Fan (2018; 2019) applies Gaussian blurring in terms of image pixelization to distort visual details. Ji et al. (2014) combines public and private data to mitigate performance degradation due to differential privacy noises. InstaHide (Huang et al., 2020) enhances privacy by mixing up input images and labels with public data.

Nevertheless, GIDs often degrade model performance (Abadi et al., 2016; Wei et al., 2021a), while private data protection techniques may fail to fully obscure sensitive information (Carlini et al., 2021;

Luo et al., 2022; Yu et al., 2025). These challenges highlight the fundamental trade-off between privacy protection and model utility. To address these challenges, we propose GradientHide, which leverages additional public data to obscure private information, including gradients, without directly mixing it with private data. An additional update step using public data is introduced before each client transmits gradients to the server, effectively masking sensitive information embedded in model parameters. Maintaining model performance requires aligning the label distribution of public data with that of private data, which we achieve through CLIP's zero-shot inference, enabling effective training despite distributional gaps.

Our contributions are as follows:

- We show that strategically injecting public data into the training pipeline can effectively obscure private information in gradients, a direction that has not been thoroughly explored in prior FL privacy defenses.
- We demonstrate that zero-shot semantic alignment via CLIP enables effective use of public data without incurring additional training costs. This design enables successful training across datasets with large distributional gaps, without compromising model performance.
- We extensively evaluate GradientHide against state-of-the-art GIAs together with quantifying information leakage using mutual information, showing it best achieves the trade-off between privacy defense and model utility over the existing GIDs. Moreover, GradientHide yields semantically shifted yet visually coherent reconstructions, misleading attackers into overestimating inversion success.

## 2 METHOD: GRADIENTHIDE

### 2.1 THREAT MODEL

In this paper, following prior works (Aji & Heafield, 2017; Fang et al., 2023; Geiping et al., 2020; Jeon et al., 2021; Li et al., 2022; Sun et al., 2021; Yin et al., 2021; Zhang et al., 2025), we consider a federated learning setting involving clients and an *honest-but-curious* server. The server serves as the adversary, which seeks to infer clients' private data without ever obtaining direct access. It is granted full knowledge of the model architecture and the local updated model parameters transmitted at each round, yet it cannot tamper with either the model or the gradient updates. Moreover, we assume the adversary can leverage external information, such as publicly available datasets and pre-trained neural networks (*e.g.*, GAN-based generators), to bolster its reconstruction attacks. On the defense side, each benign client applies local privacy mechanisms and retains only the global model parameters released at each round. Clients have no visibility into the adversary's specific techniques or configuration. Finally, to assess the strongest possible threat, our evaluations were conducted under the most adversarially favorable conditions, *i.e.*, the attacker's optimal configuration was adopted when assigning the batch size for each client, and clients' gradient updates were transmitted after training on a local batch. Moreover, we impose only minimal restrictions on the adversary's computational resources and memory.

### 2.2 PROBLEM FORMULATION

Let $f_\theta$ be a classifier parametrized by $\theta$. Given a pair of data sample $(\mathbf{x}^*, \mathbf{y}) \in \mathbb{R}^{H \times W \times L} \times \{0,1\}^C$, where $H$, $W$, and $L$ denote the height, width, and number of channels of an image, respectively, and $C$ is the number of classes, the client uploads the gradient $g$ computed as $\nabla \ell (f_\theta(\mathbf{x}^*), \mathbf{y})$.

The reconstruction of $\mathbf{x}^*$ via a gradient inversion attack can be formulated as the following optimization problem:

$$\hat{\mathbf{x}} = \operatorname*{argmin}_{\mathbf{x} \in \mathbb{R}^{H \times W \times L}} \mathcal{L} \left( \nabla \ell \left( f_\theta \left( \mathbf{x} \right), \mathbf{y} \right), g \right) + \lambda \mathcal{R} \left( \mathbf{x} \right), \tag{1}$$

where $\mathcal{L}$ is a distance function for measuring the discrepancy between $g$ and the gradient computed by $\mathbf{x}$, and $\mathcal{R}$ is a regularization term.

In optimization-based attacks (Dimitrov et al., 2024; Geiping et al., 2020; Yin et al., 2021; Zhao et al., 2020; Zhu et al., 2019), the objective function is expressed as in Eq. (1). Since $g$ encodes information about both the model parameters and the label corresponding to $\mathbf{x}^*$, an attacker can

Figure 1: Workflow of GradientHide. (1) The model is first updated using private data following the FedAvg procedure. Transmitting the model at this stage without any defense poses a risk of privacy leakage. (2) A batch data of size $M$ is randomly sampled from an aligned public dataset. (3) An additional update is performed on the aligned public data before the model parameters are transmitted, thereby mitigating privacy risks.

iteratively refine a dummy input $\mathbf{x}$ through optimization to obtain $\hat{\mathbf{x}}$, which approximates $\mathbf{x}^*$. To improve reconstruction quality, (Fang et al., 2023; Hatamizadeh et al., 2022; Jeon et al., 2021; Li et al., 2022) adopt a generative prior by expressing $\mathbf{x}$ as $G(\mathbf{z})$, where $G$ is a pre-trained generator and $\mathbf{z}$ is a latent code, thereby constraining reconstructions to the data manifold and enhancing their visual plausibility. $G$ also may alternatively be adaptively constructed (Yu et al., 2025), encoding implicit architectural priors without relying on pre-training.

To defend against the GIAs, defense mechanisms are applied on the client side and can be formalized as the transformation of transmitted gradients as follows:

$$\hat{g} = \mathcal{T}(g) + \varepsilon, \tag{2}$$

where $\mathcal{T}$ denotes a lossy transformation, such as compression, sparsification, clipping, or orthogonal, and $\varepsilon$ represents additive noise, *e.g.*, from differential privacy (Geyer et al., 2017; Yuan et al., 2023; Li et al., 2024). These operations jointly mitigate the risk of input reconstruction. However, these defenses (Aji & Heafield, 2017; Geyer et al., 2017; Sun et al., 2021; Wei et al., 2021b; Yuan et al., 2023; Zhang et al., 2025) often entail a trade-off between model utility and privacy protection, especially under data heterogeneity.

To address this challenge, we propose GradientHide, which maintains model performance while offering effective protection against gradient inversion attacks. As described in Sec. 3.3, it achieves stable accuracy even under challenging data heterogeneity, demonstrating its practicality and robustness in real-world federated learning. Related work on GIAs and GIDs is summarized in Sec. A of Appendix.

## 2.3 PROPOSED FRAMEWORK

GradientHide aims to prevent private data from being leaked via transmitting gradients in FL by introducing an additional update step using public data to obfuscate sensitive information. This mechanism can be naturally integrated into FL training routines, as illustrated in Fig. 1. In Sec. 2.3.1, we first introduce Label Alignment, which is performed on each client before FL training to facilitate additional updates, as illustrated in Fig. 7 of Appendix B. We then detail the procedure for performing these updates in each communication round. The pseudocode of the proposed algorithm is depicted in Algorithm 1 in Sec. C of Appendix. To evaluate GradientHide's privacy protection, we quantify information leakage using mutual information in Sec. 2.3.2. The derivations will be deliberately described in Sec. D of Appendix.

### 2.3.1 TWO-STAGE LOCAL UPDATE IN FEDERATED LEARNING

Consider a federated learning system with $N$ clients, where each client $i$ ($1 \leq i \leq N$) holds a private dataset $\mathcal{D}_i = \{(\mathbf{x}_j^i, \mathbf{y}_j^i)\}_{j=1}^B$ and a public dataset $\mathcal{H}_i = \{(\tilde{\mathbf{x}}_j^i, \tilde{\mathbf{y}}_j^i)\}_{j=1}^S$. Here, $\mathbf{x}_j^i$ and $\tilde{\mathbf{x}}_j^i$ denote the $j$-th input image of client $i$ from the private and public datasets, respectively, while $\mathbf{y}_j^i$ and $\tilde{\mathbf{y}}_j^i$ are the corresponding labels. To ensure label compatibility and enable seamless integration into the federated training process, the public dataset $\mathcal{H}_i$ undergoes a Label Alignment procedure prior to the start

of federated training. Specifically, we adopt CLIP's zero-shot inference framework (Radford et al., 2021), which maps both image and text into a shared embedding space for similarity comparison.

To perform the Label Alignment, we consider $P$ different prompt templates[1], each providing a set of $C$ class-specific textual prompts, where $C$ corresponds to the number of classes in the private dataset $\mathcal{D}_i$. Formally, we denote $t_c^{(p)}$ as the prompt corresponding to the $c$-th class under the $p$-th template. An image $\tilde{\mathbf{x}}_j^i$ of the public dataset and each prompt $t_c^{(p)}$ are independently mapped into a shared embedding space by the image encoder $E_I$ and text encoder $E_T$, respectively. For each template $t_c^{(p)}$ and image $\tilde{\mathbf{x}}_j^i$, we compute the score between them by

$$s(\tilde{\mathbf{x}}_j^i, t_c^{(p)}) = \frac{\langle E_I(\tilde{\mathbf{x}}_j^i), E_T(t_c^{(p)}) \rangle}{\|E_I(\tilde{\mathbf{x}}_j^i)\| \|E_T(t_c^{(p)})\|} \tag{3}$$

to specify their alignment.

To leverage the diversity across prompt templates, we average the scores, defined in Eq. (3), over all templates. The aligned label $\hat{\mathbf{y}}$ is obtained by selecting the class with the highest mean similarity:

$$\hat{\mathbf{y}} = \underset{c}{\mathrm{argmax}} \ \frac{1}{P} \sum_{p=1}^{P} s(\tilde{\mathbf{x}}_j^i, t_c^{(p)}). \tag{4}$$

Thus, the images from the public dataset are assigned semantic labels according to Eq. (4) to align with the class distribution of private dataset, without requiring additional training or learnable components. In this process, the original labels $\{\tilde{\mathbf{y}}_j^i\}$ are discarded and replaced with new labels $\{\hat{\mathbf{y}}_j^i\}$, resulting in the aligned public dataset $\hat{\mathcal{H}}_i = \{(\tilde{\mathbf{x}}_j^i, \hat{\mathbf{y}}_j^i)\}_{j=1}^{S}$.

With the aligned datasets in place, federated training begins by having the server initialize the global model parameters $\theta^0$. In each communication round $\tau$, a subset of $K$ clients (where $K < N$) is selected to participate and download the model parameters $\theta^\tau$ from the server to conduct local update by:

$$\theta_{\mathcal{D}_i}^\tau = \underset{\theta^\tau}{\mathrm{argmin}} \frac{1}{B} \sum_{j=1}^{B} \ell\left(f_{\theta^\tau}(\mathbf{x}_j^i), \mathbf{y}_j^i\right). \tag{5}$$

This local update procedure resembles the training phase of FedAvg (McMahan et al., 2017), where each involved client $i$ independently optimizes its local model using private dataset $D_i$.

However, directly transmitting these locally updated parameters $\theta_{\mathcal{D}_i}^\tau$ can lead to privacy leakage, as they may inadvertently reveal information about the underlying private dataset $\mathcal{D}_i$. To address this concern, we introduce an additional update step using an aligned public dataset $\hat{\mathcal{H}}_i$ designed to prevent privacy leakage. This aligned public dataset $\hat{\mathcal{H}}_i$ ensures label compatibility with the private data $\mathcal{D}_i$, making it feasible for additional update steps while maintaining the model performance as possible.

Specifically, each client $i$ randomly draws a set of $M = \alpha B$ data samples, denoted by $\mathcal{M} = \{(\mathbf{x}_j, \mathbf{y}_j)\}_{j=1}^{M}$, from $\hat{\mathcal{H}}_i$, where $\alpha \in (0, 1]$ denotes the ratio of the subset size. The introduced randomness is beneficial to reduce information leakage, as will be described in Sec. 2.3.2. The client then computes a perturbation term, defined as the scaled negative gradient over this subset with learning rate $\eta$, as:

$$\boldsymbol{\xi}_{\mathcal{M}} = -\eta \left[ \frac{1}{M} \sum_{j=1}^{M} \nabla_\theta \ell\left(f_{\theta_{\mathcal{D}_i}^\tau}(\mathbf{x}_j), \mathbf{y}_j\right) \right]. \tag{6}$$

Using this perturbation term, the client forms the local model parameters to be transmitted:

$$\theta_i^\tau = \theta_{\mathcal{D}_i}^\tau + \lambda_\eta \boldsymbol{\xi}_{\mathcal{M}}. \tag{7}$$

The update in Eq. (7) is simply a gradient descent step with the scaled learning rate $\lambda_\eta \cdot \eta$, where $\lambda_\eta$ is a scaling factor. However, in the context of GradientHide, it is interpreted as a gradient-based perturbation, as its role is to introduce obfuscation rather than to optimize model performance.

---

[1]Following CLIP (Radford et al., 2021), we used 80 prompt templates. (See Appendix B.)

These model parameters $\theta_i^\tau$ with sensitive information obfuscated are actually transmitted to the central server, which aggregates them from the $K$ selected clients to update the global model:

$$\theta^{\tau+1} = \frac{1}{K} \sum_{i=1}^{K} \theta_i^\tau. \tag{8}$$

The updated global parameters $\theta^{\tau+1}$ in Eq. (8) are then broadcast to the next set of selected clients in the subsequent communication round.

### 2.3.2 ANALYZING THE INFORMATION LEAKAGE OF GRADIENTHIDE

Following the framework in (Tan et al., 2024), we formalize local training in FL as an information flow system, where the global received model $\theta^\tau$, the local updates driven by private data $\mathcal{D}_i$, and the transmitted model $\theta_{\mathcal{D}_i}^\tau$ at client $i$ constitute the ingress, internal, and egress flows, respectively. The information leakage of private data is then captured by the increase in mutual information when the server observes $\theta_{\mathcal{D}_i}^\tau$, formally quantified as $I(\mathcal{D}_i; \theta_{\mathcal{D}_i}^\tau \mid \theta^\tau)$.

With GradientHide, each client modifies the update in Eq. (5) by adding the perturbation $\boldsymbol{\xi}_{\mathcal{M}}$ defined in Eq. (6) and scaled by $\lambda_\eta$, leading to Eq. (7). Accordingly, the information leakage in this case is quantified by $I(\mathcal{D}_i; \theta_{\mathcal{D}_i}^\tau + \lambda_\eta \boldsymbol{\xi}_{\mathcal{M}} \mid \theta^\tau)$, and under the conditional independence assumption, GradientHide guarantees

$$I\left(\mathcal{D}_i; \theta_{\mathcal{D}_i}^\tau + \lambda_\eta \boldsymbol{\xi}_{\mathcal{M}} \mid \theta^\tau\right) \leq I\left(\mathcal{D}_i; \theta_{\mathcal{D}_i}^\tau \mid \theta^\tau\right). \tag{9}$$

To further examine why the leakage decreases, we rewrite the mutual information in its entropy form:

$$I(\mathcal{D}_i; \theta_{\mathcal{D}_i}^\tau + \lambda_\eta \boldsymbol{\xi}_{\mathcal{M}} \mid \theta^\tau) = h(\theta_{\mathcal{D}_i}^\tau + \lambda_\eta \boldsymbol{\xi}_{\mathcal{M}} \mid \theta^\tau) - H(\lambda_\eta \boldsymbol{\xi}_{\mathcal{M}} \mid \theta^\tau), \tag{10}$$

where the first term captures the total uncertainty of the perturbed update and the second term isolates the randomness from the perturbation. In practice, the variance of $\boldsymbol{\xi}_{\mathcal{M}}$ primarily comes from the randomness of sampling the subset $\mathcal{M}$. This randomness amplifies the conditional entropy, thereby reinforcing the reduction in information leakage.

Overall, GradientHide ensures that the transmitted updates carry strictly less exploitable information about private data, thereby improving resistance against gradient inversion attacks. More detailed derivations are provided in Sec. D of the Appendix.

## 3 EXPERIMENTS

This section presents a comprehensive empirical evaluation of our proposed defense, GradientHide. Experimental setup is described in Sec. 3.1. Then, the trade-off between privacy preserving and model performance is studied. Specifically, in Sec. 3.2, we evaluate the privacy protection of GradientHide against five representative GIAs and compare it with seven state-of-the-art GIDs. In Sec. 3.3, we examine model performance under challenging data heterogeneity, highlighting each GID's ability to sustain performance in practical federated learning environments. In Sec. 3.4, we conduct ablation studies to assess the impact of design components and hyperparameters.

### 3.1 EXPERIMENTAL SETUP

#### 3.1.1 CONFIGURATION OF GRADIENT INVERSION ATTACKS AND DEFENSES

Our evaluation covers three benchmark datasets, including CIFAR-10 (Krizhevsky et al., 2009), ImageNet (Deng et al., 2009), and FFHQ (Karras et al., 2019; Or-El et al., 2020) as the private datasets, where their corresponding public datasets are ImageNet, CIFAR-10, and UTKFace (Zhang et al., 2017), with full preprocessing details provided in Sec. F of the Appendix. The arrangement of such dataset pair is to explore the following three scenarios:

- First scenario (Label reduction): Semantically abundant labels are mapped to smaller, more abstract generic ones. In this case, we set CIFAR-10 and ImageNet as the private and public datasets, respectively.

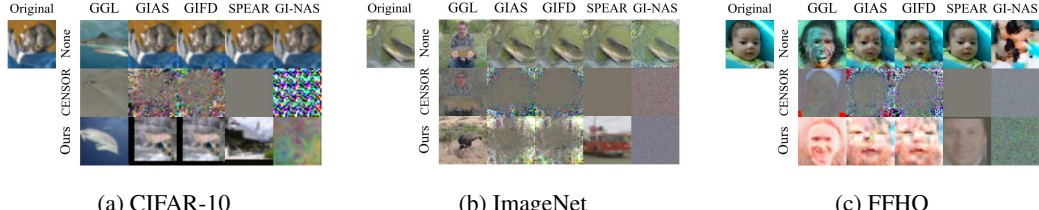

Figure 2: Reconstruction result for the first validation image (at index 0) across five GIAs and two GIDs (Ours and CENSOR (Zhang et al., 2025)) under three different private datasets. For our method, the adopted public datasets are (a) ImageNet, (b) CIFAR-10, and (c) UTKFace.

- Second scenario (Label mismatch): ImageNet and CIFAR-10 were set as the private and public datasets, respectively. Since ImageNet contains categories semantically unrelated to those in CIFAR-10 (e.g., tailed frog, sea snake, toaster), this leads to substantial inter-class semantic mismatch.

- Third scenario (Label consistency): Although Label Alignment is not necessary for FFHQ as private and UTKFace as public, we include it for completeness.

We conducted evaluations on five GIAs, including four GAN-based methods (*i.e.*, GGL (Li et al., 2022), GIAS (Jeon et al., 2021), GIFD (Fang et al., 2023), and GI-NAS (Yu et al., 2025)), and SPEAR (Dimitrov et al., 2024). The selection of GAN-based methods as the GIA baselines is based on prior studies (Li et al., 2022; Zhang et al., 2025; Yu et al., 2025) that the attack methods employing GANs (Goodfellow et al., 2014) generally surpass optimization-based ones in terms of reconstruction fidelity and attack effectiveness. Furthermore, SPEAR (Dimitrov et al., 2024) is specifically selected for its ability to *exactly* reconstruct batches of images with batch size greater than one, as evidenced by the provided PSNR results.

Following the standard experimental setup from prior works (Fang et al., 2023; Zhang et al., 2025), we configured GGL, GIAS, and GIFD with a batch size of $B = 1$, using pre-trained BigGAN (Brock et al., 2019) for CIFAR-10 (Krizhevsky et al., 2009) and ImageNet (Deng et al., 2009), and Style-GAN2 (Karras et al., 2019) for FFHQ (Karras et al., 2019). For GI-NAS, we followed its original configuration and set the minimum batch size to $B = 4$. For SPEAR, it has been shown to *exactly* reconstruct images without defenses for batch sizes up to $B = 125$. However, we observed that its reconstruction performance degrades substantially when defenses are applied and $B < 10$. Based on these findings, we set $B = 10$ in our evaluation to strike a balance between stability and attack performance. For the experiments, the federated learning model is a randomly initialized ResNet-18, except for SPEAR, which requires its own model due to methodological constraints.

To investigate the superiority of the proposed method, we compared GradientHide with the following seven GIDs: Noise (Geyer et al., 2017), Clipping (Wei et al., 2021b), Sparsi (Aji & Heafield, 2017), Soteria (Sun et al., 2021), AVP (Yuan et al., 2023), ANP (Li et al., 2024), and CENSOR (Zhang et al., 2025). We adopted the implementation provided by CENSOR (Zhang et al., 2025) and strictly followed all configurations and hyperparameters it specifies. This setup ensures fair, reproducible evaluations of both GIAs and GIDs. The default parameters for GradientHide are empirically determined to be $\lambda_\eta = 0.01$ and $\alpha = 1$ based on our ablation study in Sec. 3.4. Further details regarding hardware resources and computational costs are provided in Sec. K of the Appendix.

### 3.1.2 EVALUATION METRICS

To assess the similarity between the reconstructed and original images, we employed several quantitative metrics commonly used in image quality assessment. First, the Learned Perceptual Image Patch Similarity (LPIPS) (Zhang et al., 2018b) was used to measure the perceptual difference by comparing deep feature representations extracted from a pre-trained VGG (Simonyan & Zisserman, 2014) network. Higher LPIPS values indicate greater perceptual dissimilarity within the deep feature space. Secondly, we used the Peak Signal-to-Noise Ratio (PSNR) to quantify pixel-level fidelity by computing the logarithmic ratio between the maximum possible signal power and the power of distortion noise, where lower values indicate greater differences. Finally, the Structural Similarity

Table 1: Quantitative evaluation of reconstruction quality under four GIAs and five GIDs. Metrics are annotated ↑ to indicate that higher values are better and ↓ to indicate that lower values are preferable. For Ours, the adopted public datasets are ImageNet, CIFAR-10, and UTKFace, respectively.

| Dataset | Defense | GGL | | | GIAS | | | GIFD | | | SPEAR | | | GI-NAS | | |
|---|---|---|---|---|---|---|---|---|---|---|---|---|---|---|---|---|
| | | LPIPS↑ | PSNR↓ | SSIM↓ | LPIPS↑ | PSNR↓ | SSIM↓ | LPIPS↑ | PSNR↓ | SSIM↓ | LPIPS↑ | PSNR↓ | SSIM↓ | LPIPS↑ | PSNR↓ | SSIM↓ |
| CIFAR-10 | None | 0.5804 | 13.1796 | 0.1530 | 0.0756 | 29.0828 | 0.8814 | 0.1120 | 28.9376 | 0.8886 | 1.5E-13 | 135.2062 | 1.0000 | 0.0540 | 29.3073 | 0.9054 |
| | Noise | 0.5942 | 13.2849 | 0.1590 | 0.0927 | 28.4165 | 0.8672 | 0.0657 | 29.3635 | 0.8928 | **0.5937** | 12.5081 | 0.1114 | 0.7012 | 11.5743 | 0.1433 |
| | Clipping | 0.5896 | 12.7224 | 0.1761 | 0.0795 | 28.5682 | 0.8772 | 0.0528 | 30.2146 | 0.9067 | 1.5E-13 | 135.2062 | 1.0000 | 0.0698 | 28.4628 | 0.9045 |
| | Sparsi | 0.5862 | 13.2723 | 0.1624 | 0.0835 | 28.7247 | 0.8776 | 0.0559 | 30.1322 | 0.9044 | **0.5937** | 12.5081 | 0.1114 | 0.0066 | 31.3063 | 0.9932 |
| | Soteria | 0.6006 | 12.9742 | 0.1687 | 0.0818 | 28.2088 | 0.8710 | 0.0569 | 29.8077 | 0.8994 | 1.5E-13 | 135.2062 | 1.0000 | 0.0387 | 29.9047 | 0.9490 |
| | AVP | 0.5894 | 13.0637 | 0.1504 | 0.0944 | 27.8386 | 0.8625 | 0.0730 | 28.7186 | 0.8857 | **0.5937** | 12.5081 | 0.1114 | 0.5992 | 12.9381 | 0.4178 |
| | ANP | 0.6034 | 12.9158 | 0.1518 | **0.7068** | 12.0258 | 0.0417 | **0.7034** | 12.1962 | 0.0403 | **0.5937** | 12.5081 | 0.1114 | 0.7313 | **10.3906** | **0.0242** |
| | CENSOR | **0.6683** | 13.1665 | **0.0217** | 0.6777 | 10.9001 | **0.0366** | 0.6763 | 11.0474 | **0.0244** | **0.5937** | 12.5081 | 0.1114 | **0.7436** | 11.3016 | 0.0476 |
| | Ours | 0.6616 | **10.2322** | 0.1425 | 0.6570 | **8.6391** | 0.0907 | 0.6718 | **9.6009** | 0.0975 | 0.2214 | **9.9565** | **0.0509** | 0.7363 | 11.5716 | 0.0495 |
| ImageNet | None | 0.6264 | 13.4640 | 0.0701 | 0.4735 | 19.3213 | 0.3481 | 0.3937 | 20.9633 | 0.4417 | 1.6E-13 | 136.9422 | 1.0000 | 0.5176 | 18.3903 | 0.4766 |
| | Noise | 0.6040 | 13.6787 | 0.0754 | 0.5356 | 18.2985 | 0.2817 | 0.4909 | 18.8306 | 0.2987 | **0.5728** | 14.9550 | 0.1657 | 0.7919 | 10.6463 | 0.0361 |
| | Clipping | 0.6006 | 13.5417 | 0.0680 | 0.4500 | 19.3260 | 0.3614 | 0.4132 | 20.3635 | 0.4199 | 1.6E-13 | 136.9422 | 1.0000 | 0.4744 | 19.8168 | 0.5437 |
| | Sparsi | 0.6305 | 12.8973 | 0.0726 | 0.4486 | 19.7249 | 0.3854 | 0.3854 | 20.9803 | 0.4390 | **0.5728** | 14.9550 | 0.1657 | 0.5618 | 17.5092 | 0.4257 |
| | Soteria | 0.6118 | 13.4784 | 0.0676 | 0.4731 | 19.0028 | 0.3527 | 0.4336 | 19.9069 | 0.4003 | 1.6E-13 | 136.9422 | 1.0000 | 0.5272 | 18.2130 | 0.4458 |
| | AVP | 0.6078 | 13.5668 | 0.0714 | 0.5205 | 17.8840 | 0.2745 | 0.4575 | 19.6928 | 0.3731 | **0.5728** | 14.9550 | 0.1657 | **0.8041** | 10.4849 | 0.0388 |
| | ANP | 0.5948 | 13.3571 | 0.0702 | 0.7463 | 13.4871 | 0.0209 | 0.7515 | 13.9993 | 0.0199 | **0.5728** | 14.9550 | 0.1657 | 0.8037 | **10.1159** | **0.0302** |
| | CENSOR | **0.7716** | 14.0660 | **0.0100** | 0.7571 | 12.4095 | **0.0119** | 0.7569 | 13.3891 | **0.0088** | **0.5728** | 14.9550 | 0.1657 | 0.7602 | 11.7813 | 0.0687 |
| | Ours | 0.7193 | **11.8858** | 0.0623 | **0.7582** | 10.5030 | 0.0247 | **0.7581** | 11.2880 | 0.0520 | 0.2129 | **12.2321** | **0.1032** | 0.7772 | 11.6498 | 0.0803 |
| FFHQ | None | 0.4654 | 13.6471 | 0.2108 | 0.3854 | 16.9409 | 0.5004 | 0.4500 | 18.6836 | 0.4794 | 1.7E-13 | 138.1643 | 1.0000 | 0.7146 | 10.2093 | 0.3573 |
| | Noise | 0.4536 | 13.9732 | 0.2373 | 0.5891 | 16.3005 | 0.3341 | 0.5556 | 17.3521 | 0.3695 | **0.4923** | 18.2208 | 0.2653 | 0.7884 | 10.1938 | 0.2585 |
| | Clipping | 0.4236 | 13.8894 | 0.2434 | 0.4123 | 18.4344 | 0.4843 | 0.3961 | 20.3410 | 0.5475 | 1.7E-13 | 138.1643 | 1.0000 | 0.7040 | 10.3160 | 0.3604 |
| | Sparsi | 0.4547 | 14.0452 | 0.2293 | 0.3933 | 18.7583 | 0.5041 | 0.4198 | 20.0461 | 0.5225 | **0.4923** | 18.2208 | 0.2653 | 0.7052 | 10.3531 | 0.3670 |
| | Soteria | 0.4427 | 14.1850 | 0.2498 | 0.4299 | 18.4120 | 0.4740 | 0.4166 | 20.4566 | 0.5311 | 1.7E-13 | 138.1643 | 1.0000 | 0.7197 | 9.9597 | 0.3474 |
| | AVP | 0.4395 | 14.4959 | 0.2484 | 0.5912 | 16.4677 | 0.3340 | 0.5301 | 17.8701 | 0.4117 | **0.4923** | 18.2208 | 0.2653 | 0.7749 | **9.9387** | 0.2789 |
| | ANP | 0.4437 | 14.1618 | 0.2332 | 0.5732 | 17.1191 | 0.3601 | 0.4996 | 17.8720 | 0.4243 | **0.4923** | 18.2208 | 0.2653 | **0.8570** | 10.4834 | 0.1396 |
| | CENSOR | **0.7729** | 11.4866 | **0.0230** | **0.7894** | 9.6989 | **0.0261** | **0.8081** | 9.6716 | **0.0201** | **0.4923** | 18.2208 | 0.2653 | 0.8558 | 11.2630 | **0.1024** |
| | Ours | 0.6517 | **8.7669** | 0.1017 | 0.7081 | **9.4842** | 0.0696 | 0.7464 | **9.4453** | 0.0659 | 0.1447 | **17.3972** | **0.2038** | 0.8480 | 11.1950 | 0.1172 |

Index Measure (SSIM) (Wang et al., 2004) was used to evaluate visual similarity by jointly considering luminance, contrast, and structural information, where an SSIM value closer to 1 indicates a greater image similarity.

## 3.2 PRIVACY PROTECTION

Due to the significant impact of random initialization on inversion outcomes, we followed prior work (Fang et al., 2023) by conducting 4 independent trials for each attack and reporting the result with the best reconstruction quality. Each attack was applied to 10 distinct images, sampled at every 1000th index from the validation dataset. The reported quantitative metrics were computed as the average over these 10 inversion instances. We evaluated the effectiveness of our proposed defense, GradientHide, against five GIAs and compared its performance with seven state-of-the-art GIDs.

As shown in Table 1, GradientHide achieves the lowest PSNR in most GIDs, indicating stronger obfuscation in the pixel space and a higher degree of deviation from the original image content. In contrast, CENSOR (Zhang et al., 2025) produces the highest LPIPS and lowest SSIM, reflecting severe distortions in both perceptual and structural domains. Its reconstructions, as shown in Fig. 2, are heavily corrupted by noise and lack recognizable structures, making the attack failure immediately apparent, not to mention the fact that it couldn't maintain model performance (see Sec. 3.3).

By comparison, GradientHide produces reconstructions that remain visually coherent but semantically shifted (Fig. 2(a)), making it difficult to associate them with the original inputs. When the public dataset contains categories unrelated to the private data (*e.g.*, toaster, sea snake in ImageNet vs. CIFAR-10), the mismatch amplifies gradient uncertainty and produces blurred reconstructions (Fig. 2(b)). Moreover, when the public dataset provides semantically consistent labels with the private data (Fig. 2(c)), reconstructions still preserve facial contours, though their appearance reflects the bias of UTKFace (Zhang et al., 2017), which predominantly contains large, centered faces with

Table 2: Reconstruction quality under adaptive attack across three scenarios, comparing implementations with EOT and without EOT.

| Scenario (Private/Public datasets) | EOT | GIFD (Fang et al., 2023) | | |
|---|---|---|---|---|
| | | LPIPS ↑ | PSNR ↓ | SSIM ↓ |
| CIFAR-10/ImageNet | w/o | 0.6635 | **8.5570** | 0.0871 |
| | w/ | **0.6982** | 11.7272 | **0.0282** |
| ImageNet/CIFAR-10 | w/o | **0.7574** | **10.5890** | 0.0301 |
| | w/ | 0.7486 | 13.7648 | **0.0153** |
| FFHQ/UTKFace | w/o | 0.7464 | **9.4453** | 0.0659 |
| | w/ | **0.8198** | 11.4655 | **0.0153** |

Table 3: Comparison of reconstruction quality via adjusting $\lambda_\eta$ across three different scenarios.

| Scenario (Private/Public datasets) | $\lambda_\eta$ | GIFD (Fang et al., 2023) | | |
|---|---|---|---|---|
| | | LPIPS ↑ | PSNR ↓ | SSIM ↓ |
| CIFAR-10/ImageNet | 1 | 0.6635 | **8.5570** | **0.0871** |
| | 0.1 | **0.6751** | 9.5596 | 0.1003 |
| | 0.01 | 0.6718 | 9.6009 | 0.0975 |
| ImageNet/CIFAR-10 | 1 | 0.7574 | **10.5890** | **0.0301** |
| | 0.1 | 0.7578 | 11.0829 | 0.0516 |
| | 0.01 | **0.7581** | 11.2880 | 0.0520 |
| FFHQ/UTKFace | 1 | 0.7464 | 9.4453 | **0.0659** |
| | 0.1 | **0.7502** | 9.3619 | 0.0787 |
| | 0.01 | 0.7296 | **9.3225** | 0.0788 |

minimal background. These results highlight that both the choice of public data and its semantic alignment with the private data critically affect reconstruction fidelity, while ensuring sufficient obfuscation to mislead attackers about inversion success.

To further evaluate the robustness of our method against adaptive attack, we combine the state-of-the-art attack, GIFD, with Expectation Over Transformation (EOT) (Athalye et al., 2018), and present the implementation details in Sec. L of the Appendix. As shown in Table 2, although EOT slightly improves PSNR, the higher SSIM and lower LPIPS indicate greater deviation from the originals.

## 3.3 MODEL PERFORMANCE

The trade-off between model performance and privacy protection is a key challenge in developing practical GIDs. To evaluate model performance with the GID configurations described in Sec. 3.1.1, we trained ResNet-18 (He et al., 2016) classifiers on CIFAR-10 (Krizhevsky et al., 2009). Furthermore, we adopted FedAvg (McMahan et al., 2017) for training and set the number of communication rounds to 800, the number of local training epochs to 5, the number of clients to 100, participating ratio is 10%. We first considered a baseline scenario, where training samples are evenly distributed across clients, simulating an Independent and Identically Distributed (i.i.d.) setting. To better reflect realistic FL conditions, we also evaluated class-heterogeneous non-i.i.d. partitions generated using a Dirichlet distribution with concentration parameter $\beta$[2].

As shown in Fig. 3, GradientHide consistently demonstrates stable performance across different levels of data heterogeneity. Under the highly heterogeneous setting ($\beta = 0.1$), only GradientHide and Clipping maintain satisfactory classification accuracy, while the others experience significant performance degradation. Notably, although CENSOR defends well by generating visually uninformative images under the reconstruction attacks discussed in Sec. 3.2, it exhibits a sharp accuracy drop of approximately 40% even under the i.i.d. setting, suggesting its limited robustness and generalizability in practical federated learning scenarios.

## 3.4 ABLATION STUDY

In this section, we present several ablation studies to examine the impact of our design components and hyperparameters. All experiments follow the settings in Sec. 3.2 and Sec. 3.3. Additional results are reported in Sec. M of Appendix.

**Effect of Label Alignment in GradientHide on GIA Reconstructions** We further study the role of Label Alignment in GradientHide by adopting an alternative alignment strategy that selects the class with the *lowest*, rather than the *highest*, average similarity in Eq. (4). Table 4 shows that selecting the *lowest*-similarity class consistently improves robustness under SPEAR. For GAN-based GIAs, however, the effect depends on the choice of public dataset. When the private dataset

---

[2]Smaller $\beta$ values produce more skewed class distributions, making client data less representative of the global distribution, a characteristic common in practical FL deployments.

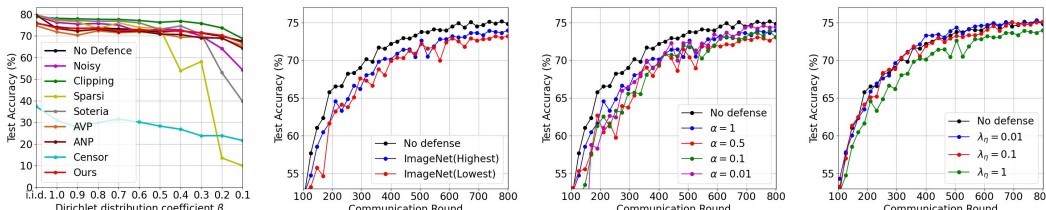

Figure 3: Model performance of GIDs under different data distributions ($\beta$).

Figure 4: Different alignment strategies ($\lambda_\eta = 1, \alpha = 1$).

Figure 5: Different ratios of public subset size $\alpha$ ($\lambda_\eta = 1$, highest alignment strategy).

Figure 6: Different scaling factors ($\alpha = 1$, highest alignment strategy).

closely matches the associated public dataset (*e.g.*, Scenario 1 (CIFAR-10/ImageNet) and Scenario 3 (FFHQ/UTKFace)), adjusting the alignment has a pronounced impact on reconstruction quality. In contrast, Scenario 2 (ImageNet/CIFAR-10) already exhibits label mismatches under the *highest* alignment strategy, which explains the instability observed under the *lowest* alignment strategy. GI-NAS is excluded from Table 4 as its reconstructions are essentially uninformative (see Fig. 2).

**Effect of GradientHide Hyperparameters on Model Performance**   To avoid the confounding effects of data heterogeneity, all these experiments were conducted under the most stable i.i.d. setting. We empirically verified the impact of different alignment strategies, different public dataset sizes, and different scaling factors. As shown in Fig. 4, the highest alignment strategy yields stable performance. Fig. 5 shows that the final accuracy is insensitive to changes in the ratio of the public subset size ($\alpha$). More details in Sec. I of the Appendix. Finally, the scaling factor $\lambda_\eta$ plays a critical role, as shown in Fig. 6, in that performance degrades when it is too large but stabilizes once $\lambda_\eta \leq 0.1$. Table 3 further shows that even at $\lambda_\eta = 0.01$, the PSNR increases marginally ($\approx 1\%$), suggesting that GIFD, the strongest attack, does not noticeably enhance reconstruction fidelity. Based on these findings, we set $\lambda_\eta = 0.01$ and used the *highest* alignment strategy as the default configuration to ensure stable performance. For privacy protection, we fixed $\alpha = 1$.

## 4   CONCLUSION & LIMITATION

We propose GradientHide, a defense mechanism against GIAs in FL that introduces an additional update step using public data after training on private data to obfuscate sensitive information in transmitted gradients. By inducing a semantic shift in the reconstructed images, despite their visual coherence, GradientHide intentionally misleads the attacker into overestimating the success of the inversion. Experimental results show that GradientHide consistently delivers strong performance, ranking among the top methods across both quantitative and qualitative benchmarks.

However, compared with standard FedAvg, the main additional cost of GradientHide comes from the one-time Label Alignment step prior to FL training, whereas the training overhead on the public subset is negligible. Developing more efficient label alignment strategies is a promising direction for future work.

Table 4: Reconstruction quality for GradientHide under two alignment strategies, three scenarios (Private/Public datasets), and four GIAs.

| Private Dataset | Alignment | GGL (Li et al., 2022) | | | GIAS (Jeon et al., 2021) | | | GIFD (Fang et al., 2023) | | | SPEAR (Dimitrov et al., 2024) | | |
|---|---|---|---|---|---|---|---|---|---|---|---|---|---|
| (Public Dataset) | Strategy | LPIPS ↑ | PSNR ↓ | SSIM ↓ | LPIPS ↑ | PSNR ↓ | SSIM ↓ | LPIPS ↑ | PSNR ↓ | SSIM ↓ | LPIPS ↑ | PSNR ↓ | SSIM ↓ |
| CIFAR-10 | highest | **0.6616** | 10.2322 | 0.1425 | 0.6570 | 8.6391 | 0.0907 | 0.6635 | 8.5570 | 0.0871 | 0.2214 | 9.9565 | 0.0509 |
| (ImageNet) | lowest | 0.6380 | **8.2945** | **0.1278** | **0.7204** | **8.0040** | **0.0431** | **0.7206** | **8.1599** | **0.0464** | **0.2655** | **8.7584** | **0.0363** |
| ImageNet | highest | **0.7193** | 11.8858 | 0.0623 | **0.7582** | **10.5030** | **0.0247** | **0.7574** | 10.5890 | **0.0301** | 0.2129 | 12.2321 | 0.1032 |
| (CIFAR-10) | lowest | 0.6771 | **10.7396** | **0.0515** | 0.7538 | 10.5776 | 0.0282 | 0.7487 | **9.3387** | 0.0790 | **0.2276** | **11.6429** | **0.0653** |
| FFHQ | highest | **0.6517** | 8.7669 | 0.1017 | 0.7081 | 9.4842 | **0.0696** | 0.7464 | 9.4453 | **0.0659** | 0.1447 | 17.3972 | 0.2038 |
| (UTKFace) | lowest | 0.6140 | 9.1112 | 0.1104 | **0.7157** | **9.3809** | 0.0835 | **0.7487** | **9.3387** | 0.0790 | **0.1685** | **16.6269** | **0.1405** |

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

APPENDIX

# A  RELATED WORKS

In this section, we review various gradient inversion attack techniques and general defense strategies designed to mitigate such threats.

## A.1  GRADIENT INVERSION ATTACKS (GIAS)

Gradient inversion attacks enable adversaries to reconstruct the private data of clients, posing a significant threat to data privacy in federated learning systems. Existing methods can be categorized into two major types: (*i*) optimization-based attacks and (*ii*) generative model-based attacks.

Optimization-based inversion attacks reconstruct training data by iteratively updating randomly initialized inputs to minimize the discrepancy between their gradients and the shared ones. Zhu et al. (2019) conceptualized this as a gradient-guided optimization process to recover private data and labels. Subsequent studies refined the approach (Zhu et al., 2019). Zhao et al. (2020) inferred labels directly from gradients to improve reconstruction accuracy. Geiping et al. (2020) achieved the inversion of higher resolution images from complex models by refining the optimization objective through an improved distance metric and an added regularization term. Yin et al. (2021) computed a registered mean image from all candidates and introduced a group consistency regularization term during optimization to reduce deviation, enabling the recovery of input images from larger batches and deep networks. Dimitrov et al. (2024) proposed SPEAR, the first algorithm to *exactly* reconstruct entire batches in FL by exploiting the low-rank structure and ReLU-induced sparsity of gradients.

Jeon et al. (2021) proposed a GAN-based approach that exploits both the latent space and the parameter space of generative models to reconstruct high-fidelity images. However, this method requires training a separate generator for each inversion task, resulting in considerable computational overhead, particularly with respect to GPU memory usage and inference time. Hatamizadeh et al. (2022) subsequently extended these attacks to Vision Transformers, demonstrating that GAN-based approaches are not limited to traditional convolutional architectures. Furthermore, Li et al. (2022) integrated generative models with label inference to improve semantic level reconstruction, ensuring better alignment between generated samples and their corresponding labels. Fang et al. (2023) introduced intermediate layer optimization,which progressively refines the input layers of the generator rather than only optimizing the latent vectors. This approach enhances the expressiveness of the generator, thereby improving the effectiveness of pre-trained GAN-based attacks in capturing underlying data distributions. More recently, Yu et al. (2025) proposed GI-NAS, which replaces reliance on domain-specific pre-trained generators by adaptively searching neural architectures to exploit implicit architectural priors, thereby achieving more generalizable and effective gradient inversion attacks.

## A.2  GRADIENT INVERSION DEFENSES (GIDS)

GIDs aim to make it difficult for attackers to reconstruct clients' private data while preserving the utility of the trained model. Under the standard FL setting, existing approaches can be categorized according to the protected target: (*i*) data protection and (*ii*) gradient protection.

In the category of data protection, MixUp (Zhang et al., 2018a) offers a simple mechanism in this direction by generating virtual samples through linear interpolation of pairs of inputs and labels. Although originally proposed as a data augmentation technique, the idea of mixing inputs has inspired several follow-up defense strategies. Pang et al. (2020) further leverage this mixing principle by combining the input with random clean samples during inference to mitigate adversarial perturbations. While their method focuses on adversarial robustness rather than privacy, it illustrates how input mixing can alter the information accessible to attackers. Building on MixUp, InstaHide (Huang et al., 2020) applies mixing more directly for privacy protection. Each private image is combined with several private and public images and then processed with a random pixel-wise mask. However, later analysis showed that the security of InstaHide critically depends on keeping the encoded inputs hidden. The attack by Carlini et al. (2021) exploits the fact that each encoded sample is nearly a linear mixture of one private image and several known public images. When the attacker observes these

encoded inputs, they can cluster them in feature space, identify the public components, and invert the mixing to reconstruct the private image. This demonstrates that defenses exposing instance-level encodings may inadvertently reveal enough information for reconstruction.

On the other hand, in gradient protection, differential privacy (DP) (Geyer et al., 2017) mitigates information leakage by perturbing gradients before transmitting them to the central server. A common approach is to add Gaussian noise to the gradients, providing rigorous theoretical privacy guarantees. However, achieving these guarantees often requires injecting substantial noise (Abadi et al., 2016; Wei et al., 2021a), which can significantly degrade the performance of the global model due to the resulting trade-off between privacy and utility. To alleviate this issue, recent works have introduced noise scheduling mechanisms, such as AVP (Yuan et al., 2023), which designs a time-varying geometric series for noise amplitudes, and ANP (Li et al., 2024), which alternately permutes noise across communication rounds to achieve near cancellation. Both approaches aim to balance privacy and utility in FL better. Gradient clipping (Wei et al., 2021b) incorporates differential privacy by limiting the influence of large gradients, thereby stabilizing training while preserving privacy guarantees. Gradient sparsification (Aji & Heafield, 2017) transmits only the components of the gradient with the largest magnitudes to reduce communication overhead. However, adversaries can still infer sensitive information from the sparse gradient patterns, rendering such methods vulnerable to gradient-based inference attacks (Fang et al., 2023; Li et al., 2022; Zhao et al., 2020). Since sparsification often involves applying a deterministic gradient mask, the proportion of non-zero entries can leak information about the underlying gradients. Soteria (Sun et al., 2021) perturbs intermediate image representations to obscure sensitive information during gradient computation while preserving the utility of the model. Maximizing the distance between the original and reconstructed images under a sparsity constraint reduces information leakage, but introduces computational overhead. CENSOR (Zhang et al., 2025) proposes a subspace-based gradient projection mechanism, in which gradients are projected onto multiple orthogonal subspaces at each layer. Among the resulting candidates, the gradient that minimizes the training loss is selected and transmitted to the server. This approach provides resilience against gradient inversion attacks while preserving the utility of the model.

## B  PROMPT DESIGN FOR LABEL ALIGNMENT

In the literature, *prompt customization* (often referred to as prompt engineering) has been shown to play a critical role in improving zero-shot performance. This phenomenon, first studied in the context of large language models such as GPT-3 (Brown et al., 2020; Gao et al., 2020), was later systematically explored by CLIP (Radford et al., 2021) in the vision-language domain. Instead of relying solely on a generic template such as "`a photo of a {class}`", CLIP introduced 80 diverse prompt templates with varied phrasing and contextual cues. Examples include "`a cropped photo of a {class}`" and "`a photo of many {class}`", which help capture richer semantic nuances for each class and thereby improve generalization across datasets.

In our setting, these prompt templates serve as the basis for *Label Alignment*, where semantic similarity between public images and private class names is computed through multiple prompt variants. The detailed procedure is illustrated in Fig. 7. Since our method directly follows CLIP's standard configuration, we verify that this established effect of template diversity also holds under our alignment setup. Specifically, we compare the default 80-template configuration with the commonly used generic prompt "`a photo of a {class}`".

When aligning ImageNet (val) images to CIFAR-10 class names, the generic-prompt variant produces noticeably less stable label assignments: roughly $20\%$ of the aligned labels differ from those obtained using the full template set. This observation aligns with CLIP's original findings, confirming that template diversity reduces estimation variance and enhances semantic reliability in our setting as well.

Importantly, the computational overhead of using all 80 templates is minimal, only about 13 seconds more than the single-prompt variant, because the dominant cost arises from CPU–GPU data transfer rather than GPU computation. Given its negligible overhead and substantially more stable alignment behavior, we adopt the CLIP 80-template design as the default configuration in our method.

Figure 7: Illustration of Label Alignment. For each public image, similarity scores are computed using prompt templates derived from each private class name. The scores for each class are averaged, and the class with the highest mean similarity is assigned as the aligned label.

## C PSEUDOCODE

In this section, we present the pseudocode of the proposed GradientHide method for defending against gradient inversion attacks. The complete algorithm is detailed in Algorithm 1.

## D ANALYZING THE INFORMATION LEAKAGE OF GRADIENTHIDE

In this section, we employ the framework proposed in (Tan et al., 2024) to analyze the information leakage of GradientHide. To make this analysis self-contained, we start by reviewing some important results in (Tan et al., 2024). Finally, we conclude this section by deriving the main result of this work.

First, we recall that the reconstruction error of gradient inversion attacks is correlated with the transmitted information.

**Theorem 1** (Lower bound for reconstruction error (Tan et al., 2024)). *For any random variable* $\mathbf{D} \in \mathbb{R}^d$ *and* $\mathbf{W} \in \mathbf{R}^m$, *we have*

$$\mathbb{E}\left[\left\|\mathbf{D} - \hat{\mathbf{D}}\right\|^2 / d\right] \geq \frac{e^{\frac{2h(\mathbf{D})}{d}}}{2\pi e} e^{\frac{-2I(\mathbf{D};\mathbf{W})}{d}}, \tag{11}$$

*where* $\hat{\mathbf{D}}$ *is a reconstruction of* $\mathbf{D}$ *by* $\mathbf{W}$, *and* $\mathbb{E}[\cdot]$, $h(\cdot)$, *and* $I(\cdot; \cdot)$ *denote the expectation, entropy, and mutual information, respectively.*

In the FL scenario, we consider $\mathbf{D}$ and $\mathbf{W}$ to be the private data (i.e., $\mathcal{D}_i$) and the transmitted parameter (i.e., $\theta_i^\tau$) at client $i$. Since $h(\mathcal{D}_i)$ is a constant, the lower bound is controlled by $I(\mathcal{D}_i; \theta_i^\tau)$, which indicates that if $\theta_i^\tau$ contains less information of $\mathcal{D}_i$, the reconstruction error is necessarily larger.

Furthermore, attackers are assumed to have access to both $\theta_i^\tau$ and $\theta^\tau$ (i.e., the transmitted parameter downloaded by the clients from the server) at each round $\tau$. Therefore, as stated in (Tan et al., 2024), $I(\mathcal{D}_i; \theta_i^R, \theta^0)$ is simplified to calculate $\sum_{\tau=0}^{R-1} I(\mathcal{D}_i; \theta_i^\tau | \theta^\tau)$.

The potential leakage of private information through communicated parameters is captured by the increase in Mutual Information (MI) when the server observes $\theta_i^\tau$. This can be formally expressed as

$$I\left(\mathcal{D}_i; \theta^\tau, \theta_i^\tau\right) = \underbrace{I\left(\mathcal{D}_i; \theta^\tau\right)}_{\text{Prior}} + \underbrace{I\left(\mathcal{D}_i; \theta_i^\tau \mid \theta^\tau\right)}_{\text{Information Leakage } (\Delta I)}. \tag{12}$$

We can further reformulate the information leakage term in Eq. (12) in terms of entropy to gain more intuitive insights:

$$I\left(\mathcal{D}_i; \theta_i^\tau \mid \theta^\tau\right) = h(\theta_i^\tau \mid \theta^\tau) - h(\theta_i^\tau \mid \mathcal{D}_i, \theta^\tau). \tag{13}$$

---

**Algorithm 1** GradientHide

---

**Input:** $R$ is the number of communication rounds, $\lambda$ is the fraction of participating clients, $N$ is the total number of clients, $\mathcal{H}_i$ is the public dataset of client $i$, $\mathcal{D}_i$ is the private dataset of client $i$, $Z$ is the local training epochs, $b$ is the local mini-batch size, $\eta$ is the learning rate.
**Output:** $\theta^{\tau+1}$ is the aggregated global model parameters at the $\tau$-th communication round.

---

**Server executes:**
  **for** each communication round $\tau = 0, 1, \ldots, R-1$ **do**
    **if** $\tau = 0$ **then**
      Initialize $\theta^0$
      **for** each client $i \in N$ **do**
        $\hat{\mathcal{H}}_i \leftarrow$ **LabelAlignment**$(\mathcal{H}_i, \mathcal{D}_i)$
      **end for**
    **end if**
    $K \leftarrow \max(\lambda \cdot N, 1)$
    $\mathcal{S}_\tau \leftarrow$ (random set of $K$ clients)
    **for** each client $i \in \mathcal{S}_\tau$ **do**
      $\theta_i^\tau \leftarrow$ **ClientUpdate**$(\theta^\tau, i)$
    **end for**
    $\theta^{\tau+1} = \frac{1}{K} \sum_{i=1}^{K} \theta_i^\tau$              *// Eq. (8)*
  **end for**
  **return** $\theta^{\tau+1}$

**ClientUpdate**$(\theta^\tau, i)$**:**
  **for** each local epoch from 1 to $Z$ **do**
    $E \leftarrow \left\lceil \frac{|\mathcal{D}_i|}{b} \right\rceil$              *// $\lceil \cdot \rceil$ is the ceiling function.*
    **for** each iteration $e$ from 1 to $E$ **do**
      $\mathcal{B} \leftarrow$ (split $\mathcal{D}_i$ into batches of size b)
      **for** the batch $\{(x_j^i, y_j^i)\}_{j=1+b\cdot(e-1)}^{b\cdot e} \in \mathcal{B}$ **do**
        $\theta_{\mathcal{D}_i}^\tau \leftarrow \theta^\tau - \eta \left( \frac{1}{b} \sum_{j=1+b\cdot(e-1)}^{b\cdot e} \nabla \ell \left( f_{\theta^\tau}(x_j^i), y_j^i \right) \right)$    *// Eq. (5)*
      **end for**
    **end for**
  **end for**
  Sample a subset $\mathcal{M} = \{(\mathbf{x}_j, \mathbf{y}_j)\}_{j=1}^{M}$ from $\hat{\mathcal{H}}_i$
  $\theta_i^\tau \leftarrow \theta^\tau - \lambda_\eta \eta \left( \frac{1}{M} \sum_{j=1}^{M} \nabla \ell \left( f_{\theta_{\mathcal{D}_i}^\tau}(\mathbf{x}_j), \mathbf{y}_j \right) \right)$     *// Eq. (6) & Eq. (7)*
  **return** $\theta_i^\tau$

**LabelAlignment**$(\mathcal{H}, \mathcal{D})$**:**
  Let $\mathcal{C} = \{c_1, c_2, \ldots, c_C\}$ be the set of class names in the private dataset $\mathcal{D}$
  Initialize $\hat{\mathcal{H}} \leftarrow \emptyset$
  **for** $(\tilde{\mathbf{x}}, \tilde{\mathbf{y}}) \in \mathcal{H}$ **do**
    $\hat{y} = \mathrm{argmax}_{c \in \mathcal{C}} \ \frac{1}{P} \sum_{p=1}^{P} s(\tilde{\mathbf{x}}, t_c^{(p)})$ *// $t_c^{(p)}$: prompt for class c under template p, cf. Eq. (4)*
    $\hat{\mathcal{H}} \leftarrow \hat{\mathcal{H}} \cup \{(\tilde{\mathbf{x}}, \hat{\mathbf{y}})\}$
  **end for**
  **return** $\hat{\mathcal{H}}$

---

Based on the update rule $\theta_i^\tau = \theta^\tau - \eta \nabla \ell(\theta^\tau, \mathcal{D}_i)$, where $\ell(\cdot)$ is the loss function, if $\theta^\tau$ is observed, $\theta_i^\tau$ becomes a deterministic function of $\mathcal{D}_i$, and thus its conditional entropy is finite. Moreover, if $\mathcal{D}_i$ and $\theta^\tau$ are both observed, $\theta_i^\tau$ is fully determined, leading to $h(\theta_i^\tau \mid \mathcal{D}_i, \theta^\tau) \to -\infty$. In this case, $I(\mathcal{D}_i; \theta_i^\tau \mid \theta^\tau) \to +\infty$, which indicates an unbounded risk of privacy leakage in the absence of protection.

To mitigate such leakage, we inject the gradient update perturbation $\boldsymbol{\xi}_{\mathcal{M}}$ computed from public data during the local update (cf. Eq. (7)). Here, a scaling factor $\lambda_\eta$ is introduced to control the strength

of the injected perturbation, which can be interpreted as a multiplicative adjustment to the learning rate. Let the client update with this perturbation be modeled as Eq. (7).

Substituting Eq. (7) into the information leakage term in Eq. (12), which is denoted by $\Delta I$, gives $\Delta I = I\left(\mathcal{D}_i; \theta^\tau_{\mathcal{D}_i} + \lambda_\eta \boldsymbol{\xi}_{\mathcal{M}} \mid \theta^\tau\right)$. To analyze $\Delta I$, we begin by establishing an upper bound using the Data Processing Inequality. Specifically, since the operation $\theta^\tau_{\mathcal{D}_i} + \lambda_\eta \boldsymbol{\xi}_{\mathcal{M}}$ is a deterministic function of the joint variable $(\theta^\tau_{\mathcal{D}_i}, \lambda_\eta \boldsymbol{\xi}_{\mathcal{M}})$, the information about $\mathcal{D}_i$ contained in the pair $(\theta^\tau_{\mathcal{D}_i}, \lambda_\eta \boldsymbol{\xi}_{\mathcal{M}})$ cannot be increased through this operation. This observation directly yields the first inequality:

$$I\left(\mathcal{D}_i; \theta^\tau_{\mathcal{D}_i} + \lambda_\eta \boldsymbol{\xi}_{\mathcal{M}} \mid \theta^\tau\right) \leq I\left(\mathcal{D}_i; \theta^\tau_{\mathcal{D}_i}, \lambda_\eta \boldsymbol{\xi}_{\mathcal{M}} \mid \theta^\tau\right). \tag{14}$$

Next, we decompose the right-hand side of Eq. (14) using the Chain Rule for Mutual Information. According to this rule, the mutual information between $\mathcal{D}_i$ and the pair $(\theta^\tau_{\mathcal{D}_i}, \lambda_\eta \boldsymbol{\xi}_{\mathcal{M}})$ can be separated into two terms: the mutual information between $\mathcal{D}_i$ and $\theta^\tau_{\mathcal{D}_i}$, and the conditional mutual information between $\mathcal{D}_i$ and $\lambda_\eta \boldsymbol{\xi}_{\mathcal{M}}$ given $\theta^\tau_{\mathcal{D}_i}$. Accordingly, we obtain the following equality:

$$I\left(\mathcal{D}_i; \theta^\tau_{\mathcal{D}_i}, \lambda_\eta \boldsymbol{\xi}_{\mathcal{M}} \mid \theta^\tau\right) = I\left(\mathcal{D}_i; \theta^\tau_{\mathcal{D}_i} \mid \theta^\tau\right) + I\left(\mathcal{D}_i; \lambda_\eta \boldsymbol{\xi}_{\mathcal{M}} \mid \theta^\tau, \theta^\tau_{\mathcal{D}_i}\right). \tag{15}$$

By combining Eq. (14) and Eq. (15), we can derive a practical upper bound for $\Delta I$:

$$\Delta I = I\left(\mathcal{D}_i; \theta^\tau_{\mathcal{D}_i} + \lambda_\eta \boldsymbol{\xi}_{\mathcal{M}} \mid \theta^\tau\right) \leq I\left(\mathcal{D}_i; \theta^\tau_{\mathcal{D}_i} \mid \theta^\tau\right) + I\left(\mathcal{D}_i; \lambda_\eta \boldsymbol{\xi}_{\mathcal{M}} \mid \theta^\tau, \theta^\tau_{\mathcal{D}_i}\right). \tag{16}$$

When analyzing the source of the perturbation term $\boldsymbol{\xi}_{\mathcal{M}}$, we note that it is generated from a random subset $\mathcal{M} \subseteq \hat{\mathcal{H}}$, and is therefore independent of the private dataset $\mathcal{D}_i$. We formalize this by assuming $\boldsymbol{\xi}_{\mathcal{M}} \perp\!\!\!\perp \mathcal{D}_i \big| (\theta^\tau, \theta^\tau_{\mathcal{D}_i})$, which directly implies that the second term in Eq. (16) vanishes, where $\perp\!\!\!\perp$ denotes conditional independence of random variables. In particular, $I\left(\mathcal{D}_i; \lambda_\eta \boldsymbol{\xi}_{\mathcal{M}} \mid \theta^\tau, \theta^\tau_{\mathcal{D}_i}\right) = 0$ and hence

$$I\left(\mathcal{D}_i; \theta^\tau_{\mathcal{D}_i} + \lambda_\eta \boldsymbol{\xi}_{\mathcal{M}} \mid \theta^\tau\right) \leq I\left(\mathcal{D}_i; \theta^\tau_{\mathcal{D}_i} \mid \theta^\tau\right). \tag{17}$$

Therefore, through Eq. (17), we can see that injecting conditionally independent perturbations will not increase information leakage.

**Remark.** The conclusion in Eq. (17) holds only if the sampling of the public subset $\mathcal{M}$ is independent of the private dataset $\mathcal{D}_i$. If the sampling procedure were to incorporate information from $\mathcal{D}_i$ (e.g., leveraging label information from the last mini-batch of $\mathcal{D}_i$ to guide the sampling in order to mitigate model corruption), the independence assumption would no longer be valid, and the perturbation $\boldsymbol{\xi}_{\mathcal{M}}$ could potentially leak additional private information.

We can further express the conditional mutual information in terms of entropy. Under the conditional independence assumption introduced above, we have

$$\begin{aligned} I(\mathcal{D}_i; \theta^\tau_{\mathcal{D}_i} + \lambda_\eta \boldsymbol{\xi}_{\mathcal{M}} \mid \theta^\tau) &= h(\theta^\tau_{\mathcal{D}_i} + \lambda_\eta \boldsymbol{\xi}_{\mathcal{M}} \mid \theta^\tau) - h(\theta^\tau_{\mathcal{D}_i} + \lambda_\eta \boldsymbol{\xi}_{\mathcal{M}} \mid \mathcal{D}_i, \theta^\tau) \\ &= h(\theta^\tau_{\mathcal{D}_i} + \lambda_\eta \boldsymbol{\xi}_{\mathcal{M}} \mid \theta^\tau) - H(\lambda_\eta \boldsymbol{\xi}_{\mathcal{M}} \mid \theta^\tau). \end{aligned} \tag{18}$$

The second equality follows from the translation invariance of differential entropy, since $\theta^\tau_{\mathcal{D}_i}$ is fixed once $(\mathcal{D}_i, \theta^\tau)$ are given. The randomness of $\boldsymbol{\xi}_{\mathcal{M}}$ stems from sampling the public subset $\mathcal{M} \subseteq \hat{\mathcal{H}}_i$ (cf. Eq. (7)).

Here, the first term quantifies the total uncertainty of the perturbed local parameter, reflecting the combined influence of $\mathcal{D}_i$ and $\mathcal{M}$, while the second term captures the intrinsic uncertainty of the perturbation itself. Leakage depends jointly on the randomness of $\mathcal{M}$, which introduces uncertainty, and the scaling factor $\lambda_\eta$, which governs the perturbation strength.

**Why perturbation reduces leakage.** Before we show why perturbation reduces leakage, we first introduce the following lemma.

**Lemma 1** (Maximum entropy distribution). *Let $\mathbf{X} \in \mathbb{R}^d$ be a continuous random variable with $\mathbb{E}[\mathbf{X}] = \mu_{\mathbf{X}}$ and $Cov(\mathbf{X}) = \Sigma_{\mathbf{X}}$. The maximum entropy distribution is the Gaussian distribution, i.e.,*

$$h(\mathbf{X}) \leq h(\mathcal{N}(\mu_{\mathbf{X}}, \Sigma_{\mathbf{X}})) = \frac{1}{2} \ln (2\pi e)^d \det(\Sigma_{\mathbf{X}}),$$

*where $\det(\cdot)$ denotes the determinant of a matrix and $\ln(\cdot)$ dnotes the natural logarithm.*

Then, we find the upper bound of $h(\theta^\tau_{\mathcal{D}_i} + \lambda_\eta \boldsymbol{\xi}_{\mathcal{M}} \mid \theta^\tau)$ in Eq. (18). Let $Cov(\theta^\tau_{\mathcal{D}_i}) = \Sigma_\theta$. For the variance of $\boldsymbol{\xi}_{\mathcal{M}}$, let $\mathbf{g}_1, \mathbf{g}_2, \ldots,$ and $\mathbf{g}_N$ be the gradients computed using the public dataset. We compute the covariance of the $\mathbf{g}_i$'s and denote it as $\Sigma_{\mathbf{g}}$. Then,

$$Cov(\boldsymbol{\xi}_{\mathcal{M}}) = \frac{\eta^2}{M^2} Cov\left(\sum_{i \in \mathcal{M}} \mathbf{g}_i\right) = \frac{\eta^2}{M^2}\left(\sum_{i \in \mathcal{M}} Cov(\mathbf{g}_i) + \sum_{i,j \in \mathcal{M}, \ i \neq j} Cov(\mathbf{g}_i, \mathbf{g}_j)\right)$$

$$= \frac{\eta^2}{M^2}\left(M\Sigma_{\mathbf{g}} + M(M-1)\left(-\frac{1}{N-1}\Sigma_{\mathbf{g}}\right)\right) = \frac{\eta^2}{M}\Sigma_{\mathbf{g}}\frac{N-M}{N-1}.$$

Therefore,

$$h(\theta^\tau_{\mathcal{D}_i} + \lambda_\eta \boldsymbol{\xi}_{\mathcal{M}} \mid \theta^\tau) \leq \frac{1}{2}\ln(2\pi e)^d \det\left(\Sigma_\theta + \lambda_\eta^2 \frac{\eta^2}{M}\Sigma_{\mathbf{g}}\frac{N-M}{N-1}\right), \tag{19}$$

which follows from Lemma 1.

The second term in Eq. (18) arises from the randomness in subset selection. Specifically, a public dataset of size $N$ allows for $\binom{N}{M}$ possible subsets of size $M$, and the larger the number of possible subsets, the higher the uncertainty introduced by the perturbation. Formally, the entropy of the sampling process is given by

$$H(\lambda_\eta \boldsymbol{\xi}_{\mathcal{M}} \mid \theta^\tau) = \lambda_\eta H(\boldsymbol{\xi}_{\mathcal{M}}) = \lambda_\eta \ln\binom{N}{M}. \tag{20}$$

According Eq. (19) and Eq. (20), we have the upper bound of Eq. (10) as

$$I(\mathcal{D}_i; \theta^\tau_{\mathcal{D}_i} + \lambda_\eta \boldsymbol{\xi}_{\mathcal{M}} \mid \theta^\tau) \leq \frac{1}{2}\ln(2\pi e)^d \det\left(\Sigma_\theta + \lambda_\eta^2 \frac{\eta^2}{M}\Sigma_{\mathbf{g}}\frac{N-M}{N-1}\right) - \lambda_\eta H(\boldsymbol{\xi}_{\mathcal{M}}). \tag{21}$$

We consider the second term of Eq. (21). This term is minimized when $M = 0$ or $M = N$, where no uncertainty arises and $H(\boldsymbol{\xi}_{\mathcal{M}}) = 0$; in contrast, it is maximized at $M = N/2$. Consequently, both the dataset size $N$ and the subset size $M$ jointly determine the perturbation randomness and govern the privacy protection, with the strongest protection attained when $M$ is close to $N/2$.

# E  TIME COST

Efficiency is crucial in federated learning, where each client performs local computation at every round before transmitting updates. Thus, it is important to quantify the overhead introduced by different GIDs to ensure that privacy protection does not impose impractical training delays.

To provide a fair comparison, we measure the per-round local computation time under the i.i.d. setting in Sec. 3.3. The reported runtime corresponds specifically to the local training pass executed before uploading gradients, since GIDs operate entirely within this stage and do not affect communication or server-side computation. All methods use the same model architecture, batch size, and hardware environment, and results are averaged over multiple rounds (mean ± standard deviation). Communication latency is excluded, as it is identical across all methods.

Table 5 reports the per-round overhead of each defense relative to the FedAvg baseline. Lightweight defenses such as Noise, Clipping, AVP, and ANP introduce minimal overhead (all < 2 seconds). In contrast, CENSOR incurs a substantial overhead of +22.66 seconds, while Soteria exceeds 700 seconds due to its iterative optimization. Our method incurs a moderate overhead of +11.64 seconds, primarily arising from a single extra forward–backward pass on a small public subset per round.

Table 5: Per-round computational time cost (seconds). "Overhead ($\Delta$s)" indicates the additional time relative to the FedAvg (None) baseline.

| Defense | None | Noise | Clipping | Sparsi | Soteria | AVP | ANP | CENSOR | Ours |
|---|---|---|---|---|---|---|---|---|---|
| Cost (s) | 8.53±0.22 | 9.1±0.16 | 9.9±0.68 | 15.29±1.29 | 719.16±2.04 | 9.96±0.33 | 10.35±0.80 | 31.19±1.85 | 20.17±0.46 |
| Overhead ($\Delta$s) | – | +0.57 | +1.37 | +6.76 | +710.63 | +1.42 | +1.82 | +22.66 | +11.64 |

Table 6: RCA (%) in reconstructed images. The lowest value in each row is highlighted in bold.

| Dataset | Attack | No Defence | Noisy | Clipping | Sparsi | Soteria | AVP | ANP | CENSOR | Ours |
|---------|--------|-----------|-------|----------|--------|---------|-----|-----|--------|------|
| CIFAR-10 | No Attack | 80 | – | – | – | – | – | – | – | – |
| | GGL | 10 | **0** | **0** | 20 | **0** | 10 | **0** | 10 | **0** |
| | GIAS | 80 | 80 | 80 | 80 | 80 | 80 | 10 | 20 | **0** |
| | GIFD | 80 | 80 | 80 | 80 | 70 | 80 | 10 | 10 | **0** |
| | SPEAR | 80 | **10** | 80 | **10** | 80 | **10** | **10** | **10** | **10** |
| | GINAS | 80 | 80 | 80 | 80 | 80 | 40 | **20** | **20** | **20** |
| ImageNet | No Attack | 30 | – | – | – | – | – | – | – | – |
| | GGL | 80 | 70 | 90 | 70 | 70 | 70 | 80 | 10 | **0** |
| | GIAS | **0** | **0** | **0** | **0** | **0** | **0** | **0** | **0** | **0** |
| | GIFD | 10 | **0** | 10 | **0** | **0** | **0** | **0** | **0** | **0** |
| | SPEAR | 30 | **0** | 10 | **0** | 10 | **0** | **0** | **0** | **0** |
| | GINAS | 30 | 10 | 30 | 30 | 30 | **0** | **0** | **0** | **0** |
| FFHQ | No Attack | 80 | – | – | – | – | – | – | – | – |
| | GGL | 50 | 30 | 30 | **10** | 20 | 20 | **10** | 20 | 20 |
| | GIAS | 40 | 30 | 40 | 60 | 40 | 20 | 40 | **0** | 20 |
| | GIFD | 40 | 40 | 50 | 50 | 40 | 50 | 40 | 20 | **0** |
| | SPEAR | 80 | **10** | 80 | **10** | 80 | **10** | **10** | 30 | **10** |
| | GINAS | **10** | 20 | **10** | 30 | 20 | **10** | **10** | 20 | 20 |

Overall, these results demonstrate that our method achieves a favorable balance between privacy, utility, and efficiency. ANP is slightly faster but provides somewhat weaker pixel-level protection than ours (Table 1). CENSOR provides strong privacy protection but exhibits the lowest utility (Fig. 3), and is nearly three times slower. In contrast, our method maintains high utility and privacy while introducing only moderate overhead, making it a practical and efficient choice for real-world FL deployments.

## F    DATASET DETAILS IN EXPERIMENTS

To ensure fair comparison, we strictly followed the preprocessing pipelines used in the official implementations of all attack baselines and defense methods. For CIFAR-10, ImageNet, and FFHQ, we directly adopted the preprocessing settings used in CENSOR. For public datasets, we matched the preprocessing settings of the corresponding private datasets to avoid introducing unintended bias.

For FFHQ, we used the officially released images at a resolution of $64 \times 64$, and adopted age labels discretized into 10-year bins, consistent with prior work. For UTKFace, we used the official $200 \times 200$ images and apply the same 10-year age binning. Importantly, we did not apply any additional face detection, cropping, or alignment to either dataset, in order to remain consistent with the original pipelines commonly used in GIA studies. Note that the two datasets naturally differ in composition: UTKFace contains tightly framed faces occupying most of the image, whereas FFHQ resembles portrait-style photos with additional background context.

## G    CAN GIDS RETAIN CLASS-SPECIFIC INFORMATION IN RECONSTRUCTED IMAGES?

While PSNR, SSIM, and LPIPS are widely used to assess reconstruction quality, these distortion-based metrics do not directly reflect the semantic information that an adversary relies on when judging whether an attack has succeeded. To evaluate how different GIDs affect the preservation of class-specific semantics in reconstructed images, we introduce a task-oriented metric based on classification utility.

Table 7: Model performance of GIDs under different data heterogeneity.

| heterogeneous setting | No Defence | Noisy | Clipping | Sparsi | Soteria | AVP | ANP | CENSOR | Ours |
|---|---|---|---|---|---|---|---|---|---|
| i.i.d. | 99.99 | 99.94 | 100.00 | 100.00 | – | 100.00 | 99.95 | 29.63 | 100.00 |
| $\beta = 0.6$ | 85.70 | 86.63 | 86.88 | 86.23 | – | 86.52 | 86.44 | 26.14 | 85.87 |
| $\beta = 0.1$ | 75.15 | 70.39 | 82.80 | 70.45 | – | 72.88 | 70.05 | – | 75.75 |

We used pre-trained ResNet-18 classifiers to infer labels from reconstructed images under different GIDs. For CIFAR-10 and FFHQ, we trained ResNet-18 from scratch for 200 epochs using SGD (batch size 256, learning rate 0.1), reaching test accuracies of $95.35\%$ and $94.67\%$, respectively. For ImageNet, we adopted the official PyTorch ResNet-18, achieving $69.76\%$ top-1 accuracy.

As shown in Table 6, GradientHide achieves the lowest RCA on CIFAR-10 and ImageNet, indicating strong suppression of class-specific cues. This behavior is consistent with the semantic drift introduced by incorporating public data during gradient perturbation. Under FFHQ, GradientHide attains the second-lowest RCA; this is expected because FFHQ and its public counterpart UTKFace share the same age-based label space, making semantic obfuscation inherently more challenging.

We also observe that GGL consistently exhibits higher RCA than other GIDs, as its latent-fitting mechanism naturally preserves semantic structure. As a result, the reconstructed images under GGL remain more class-revealing, which inadvertently assist the adversary. In contrast, GradientHide maintains low RCA across all GIAs—including GGL—demonstrating its robustness in suppressing semantic leakage regardless of the attacker's inversion strategy.

# H ADDITIONAL EVALUATION ON FFHQ MODEL PERFORMANCE

In the main paper, we prioritized CIFAR-10 for model performance evaluation, strictly adhering to common protocols in federated learning studies that analyze data heterogeneity under varying Dirichlet partitions (Fan et al., 2024; Qu et al., 2022).

To ensure our conclusions generalize across different data domains, we conducted an additional evaluation using the FFHQ dataset. The results are presented in Table 7, confirming that GradientHide maintains competitive model accuracy even on complex face data.

We omit some results from Table 7 due to practical constraints: Soteria's iterative optimization incurs substantial computational overhead (see Table 5), and in high heterogeneity settings, CENSOR's gradient orthogonalization often leads to numerical instability (e.g., gradient explosion or vanishing), preventing successful model training.

# I EFFECTS OF PUBLIC SUBSET RATIO $\alpha$ ON EFFICIENCY, UTILITY, AND PRIVACY

We first extend the runtime measurements by evaluating $\alpha \in \{1, 2, 3\}$ under the same hardware, model, and batch configurations. As shown in Table 8, the time cost increases approximately in proportion to $\alpha$. This behavior is expected because a larger $\alpha$ corresponds to a larger public subset and therefore a longer forward–backward pass during each local update. These results provide direct empirical support for our earlier claim that large $\alpha$ values increasingly introduce computational overhead.

Next, we examine the model accuracy trajectories for $\alpha \in \{0.01, 0.1, 0.5, 1, 2, 3\}$, with the results shown in Fig. 8. A consistent trend emerges: Larger $\alpha$ values introduce additional noise via the augmented public subset, causing noticeably less stable accuracy curves in the early stages of training. However, as training progresses, the trajectories gradually stabilize, and although the final accuracies remain different across $\alpha$ values, these differences becomes much smaller than those in the early training phase.

In our FL setting, only $10\%$ of clients were randomly selected in each communication round, following common practice in prior FL studies. This stochastic participation introduces additional

Table 8: Per-round computational time cost (seconds) for different $\alpha$ values. "Overhead ($\Delta$s)" indicates the additional time relative to the FedAvg (None) baseline.

| Defense | None | $\alpha = 1$ | $\alpha = 2$ | $\alpha = 3$ |
|---|---|---|---|---|
| Cost (s) | 8.53±0.22 | 20.17±0.46 | 47.60±1.15 | 66.57±3.25 |
| Overhead ($\Delta$s) | – | 11.64 | 39.07 | 58.04 |

Table 9: Comparison of reconstruction quality for different $\alpha$ values. Metrics are marked ↑ (higher is better) and ↓ (lower is better).

| $\alpha$ | GI-NAS | | |
|---|---|---|---|
| | LPIPS ↑ | PSNR ↓ | SSIM ↓ |
| 1.00 | 0.7363 | 11.5716 | 0.0495 |
| 0.75 | 0.7362 | 11.5387 | 0.0495 |
| 0.50 | 0.7312 | 11.6755 | 0.0521 |
| 0.25 | 0.7494 | 11.5716 | 0.0525 |

uncertainty in the model's optimization trajectory, which prevents the final accuracy from exhibiting a clean linear or monotonic dependence on the $\alpha$ parameter.

Finally, we assess privacy leakage under $\alpha \in \{1, 0.75, 0.5, 0.25\}$ using GI-NAS, with results summarized in Table 9. Although the computational cost grows by a factor of two or three as $\alpha$ increases (Table 8), the corresponding PSNR and other reconstruction metrics show only marginal differences. This suggests that, once the public subset reaches a sufficiently large size, further increasing $\alpha$ yields diminishing privacy benefits. In other words, privacy improvements saturate early, even as runtime continues to scale roughly linearly with $\alpha$.

Together, these new experimental results provide a comprehensive characterization of how $\alpha$ influences efficiency, model utility, and privacy leakage. They confirm that (*i*) runtime scales proportionally with $\alpha$, (*ii*) accuracy differences mainly appear during the early, unstable phase of training but diminish after convergence, and (*iii*) privacy protection remains stable across larger values of $\alpha$.

## J   CAN PUBLIC AND PRIVATE DATASETS HAVE COMPLETELY DIFFERENT SEMANTIC CONTENT AND VISUAL FEATURES?

To evaluate whether GradientHide remains effective when the public and private datasets differ substantially in both semantic content and visual characteristics, we introduce an additional setting where CIFAR-10 was used as the private dataset and FFHQ was used as the public dataset — a pair with completely disjoint semantics and appearance.

We compare this new setting against the original CIFAR-10/ImageNet pair. As shown in Table 10, using a public dataset with a larger semantic and visual gap (FFHQ) makes reconstruction no-

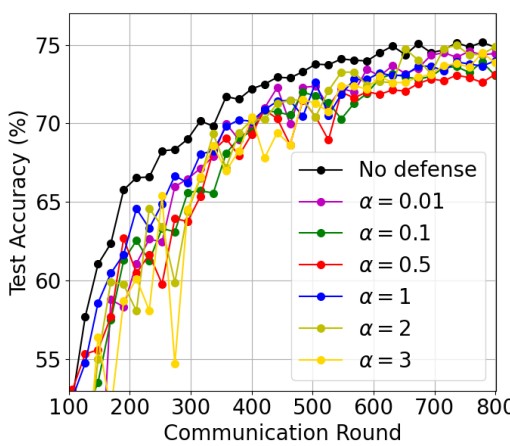

Figure 8: Different ratios of public subset size $\alpha$ ($\lambda_\eta = 1$, highest alignment strategy).

Table 10: Comparison of reconstruction quality by using the GradientHide under different public datasets (with CIFAR-10 as the private dataset). Metrics are marked ↑ (higher is better) and ↓ (lower is better).

| Scenario (Private/Public datasets) | GIFD (Fang et al., 2023) | | |
|---|---|---|---|
| | LPIPS ↑ | PSNR ↓ | SSIM ↓ |
| CIFAR-10/ImageNet | 0.6718 | 9.6009 | 0.0975 |
| CIFAR-10/FFHQ | **0.6767** | **9.5069** | **0.0877** |

Table 11: Comparison of final model accuracy under different public datasets using the GradientHide (with CIFAR-10 as the private dataset). We report results for the i.i.d. and two heterogeneous settings ($\beta = 0.6$ and $\beta = 0.1$).

| heterogeneous setting | No Defence | Public Dataset | |
|---|---|---|---|
| | | ImageNet | FFHQ |
| i.i.d. | 74.14 | 73.65 | 72.33 |
| $\beta = 0.6$ | 72.88 | 72.94 | 71.53 |
| $\beta = 0.1$ | 67.60 | 66.90 | 63.76 |

tably more difficult for GIFD, indicating that mixing with highly dissimilar public samples slightly strengthens privacy against inversion attacks.

However, this benefit comes with a slight utility cost. As reported in Table 11, model performance slightly decreases by approximately $1 \sim 3\%$ when switching from ImageNet to FFHQ as the public dataset. This highlights a privacy–utility trade-off: while a more heterogeneous public dataset can further impede reconstruction, it slightly degrades the final classification performance.

## K    DETAILS OF RESOURCES USED

All experiments were conducted on a single NVIDIA GeForce RTX 3090 GPU with 24GB of memory. In Sec. 3.2 and Sec. 3.3, each communication round introduces only an additional 10 seconds. Most of the extra time in our method comes from Label Alignment, which is performed outside FL training. On the large-scale dataset ImageNet (about 300,000 images), Label Alignment takes about 40 minutes, but it only needs to be done once.

## L    ADAPTIVE ATTACK

Given that attackers may devise adaptive strategies to circumvent defense mechanisms, we evaluate our method under adaptive attack scenarios, with detailed results reported in Sec. 3.2. Our defense introduces an additional update step that leverages public data to obscure sensitive information contained in transmitted gradients. This design reduces the identifiability of private samples from gradients and introduces further randomness, as each client independently and randomly selects a subset from the public dataset during training.

Under a white-box assumption, although the server is aware of the full content of the public dataset, it does not know which subset each client has selected. To reconstruct private data, the attacker must first sample from the public dataset and, based on these samples, estimate the targeted client's local parameters $\theta_i^\tau$, then subtract this estimate from the server model $\theta^\tau$ to obtain the private gradient $g$. However, due to the large scale of the public dataset and the inherent randomness in subset sampling, the computational and inferential complexity of reconstruction is significantly increased.

To strengthen the evaluation and counteract these stochastic effects, we combine the state-of-the-art gradient inversion method GIFD (Fang et al., 2023) with the Expectation Over Transformation (EOT) framework (Athalye et al., 2018). Specifically, the attacker performs multiple independent samplings from the public dataset, averages the resulting gradient estimates, and thereby obtains an approximation of the true private gradient. This procedure reduces the variance introduced by randomness and enhances the accuracy of subsequent gradient inversion, thus expecting to provide a more stringent and reliable evaluation of our defense.

## M    MORE ABLATION STUDIES

### M.1    ORIGIN OF THE RECONSTRUCTED IMAGE UNDER GRADIENTHIDE: PRIVATE DATASET OR NOT?

To examine whether the images reconstructed from their originals can be traced back to the private dataset, we adopt a feature-based image retrieval approach inspired by Neural Codes (Babenko et al., 2014), which shows that deep features from convolutional networks are effective descriptors even when trained on unrelated tasks. Specifically, we use a pre-trained ResNet-18 model, remove its classification head, and extract feature vectors from the penultimate layer. These features are indexed with FAISS (Douze et al., 2024) to enable efficient approximate nearest-neighbor search under the L2 distance metric.

In our evaluation, we analyze reconstructed images from four representative GIAs (GGL, GIAS, GIFD, and SPEAR) under GradientHide. We exclude GI-NAS since its reconstructed images are noise-like patterns, which provide no meaningful retrieval results. For each reconstructed image, we retrieve the nearest neighbor from the private dataset based on feature similarity.

On the CIFAR-10 and ImageNet, the retrieved neighbors differ substantially from the reconstructed images in both visual appearance and L2 distance (see Figs. 9 and 10). In contrast, on the FFHQ, the retrieved neighbor exhibits a low L2 distance to the reconstructed face but still displays clear semantic discrepancies (e.g., gender or age), highlighting that feature similarity does not guarantee perceptual or semantic alignment (see Fig. 11).

Taken together, these results suggest that GradientHide not only perturbs visual features but also disrupts semantic consistency, making reconstructed images less traceable to their true private origins.

### M.2    ADDITIONAL QUALITATIVE RESULTS

In addition to Fig. 2, we present further reconstructed images for different GIAs under GradientHide across multiple datasets (see Figs. 12– 26).

For GGL, GIAS, and GIFD, we observe distinct artifacts depending on the dataset. On CIFAR-10, reconstructions exhibit significant semantic shifts (see Figs. 12, 15, and 18). On ImageNet, they instead produce blurred results (see Figs. 13, 16, and 19). On FFHQ, where image structures remain simple, reconstructed samples still show attribute shifts such as gender or age (see Figs. 14, 17, and 20).

For SPEAR, we observe semantic shifts on CIFAR-10 and ImageNet (see Figs. 21, and 22) and attribute shifts on FFHQ (see Fig. 23). For GI-NAS, reconstructions are heavily blurred across all datasets (see Figs. 24, 25, and 26).

Overall, GradientHide introduces perturbations derived from public data that preserve coarse visual coherence while inducing semantic deviations. Such misalignment causes attackers to overestimate inversion quality, even though the reconstructions diverge from the true content.

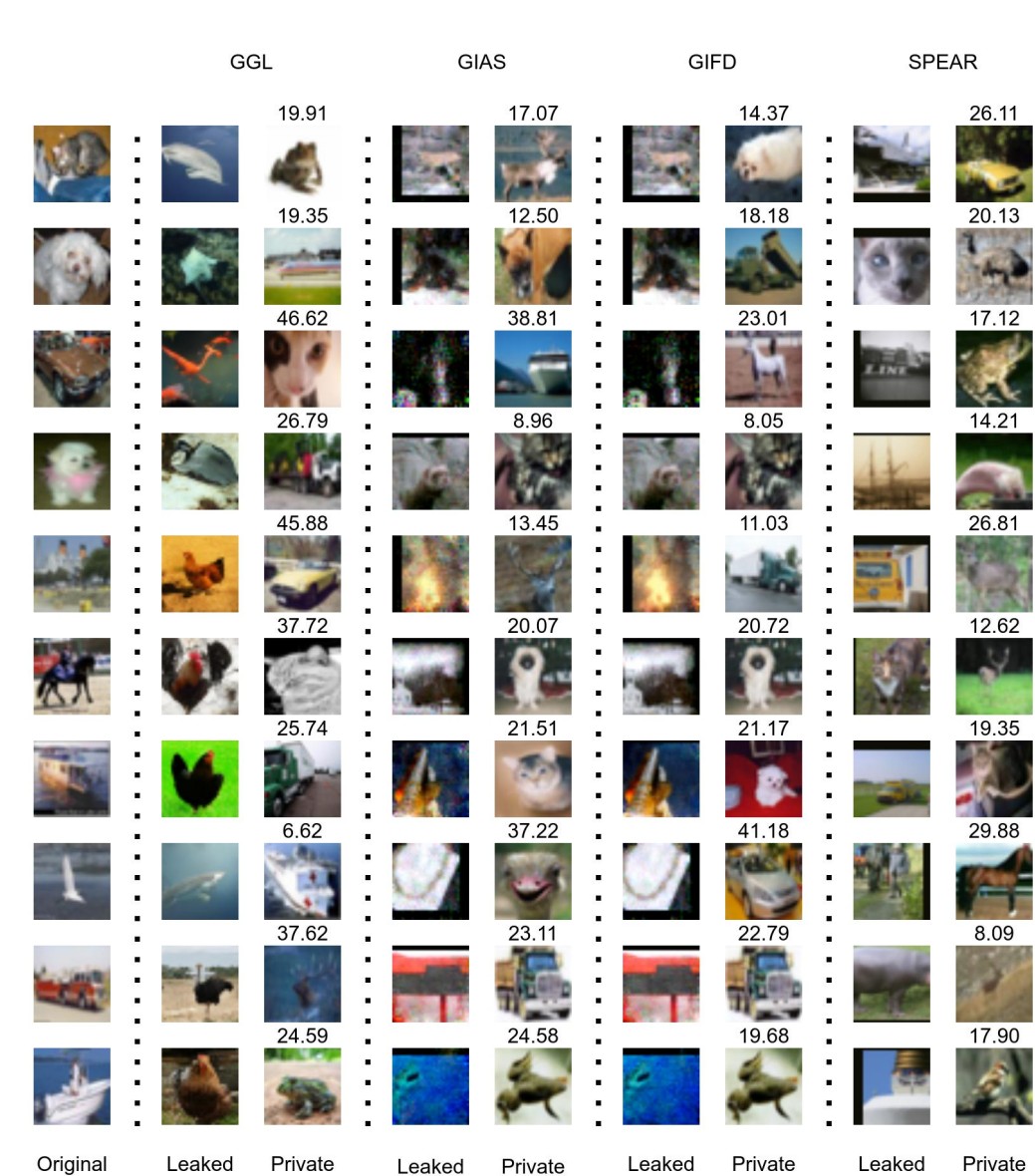

Figure 9: Reconstructed (leaked) CIFAR-10 images under GradientHide using GIAs. The L2 distance between each reconstruction and its nearest retrieved private image is shown above the retrieved image.

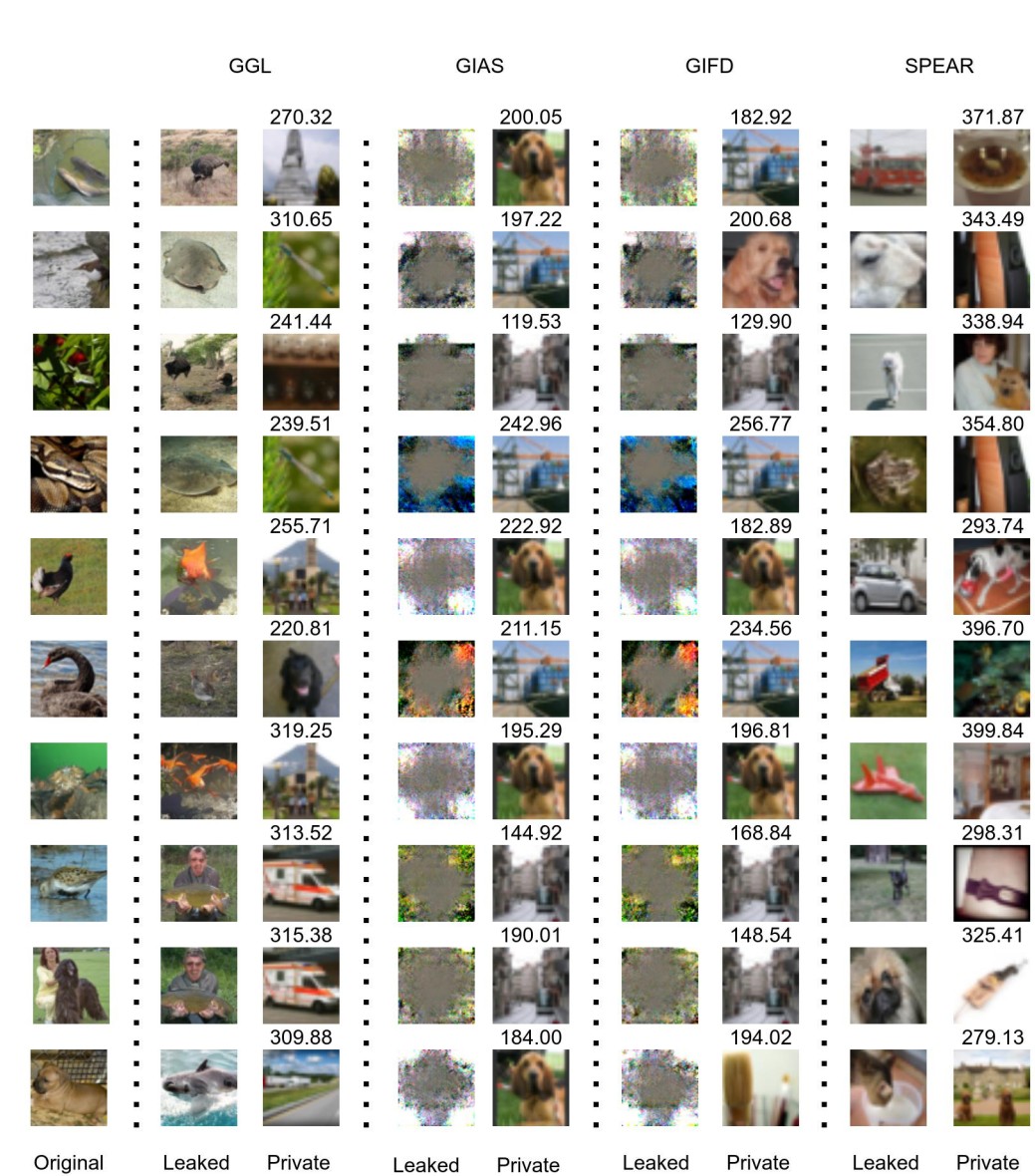

Figure 10: Reconstructed (leaked) ImageNet images under GradientHide using GIAs. The L2 distance between each reconstruction and its nearest retrieved private image is shown above the retrieved image.

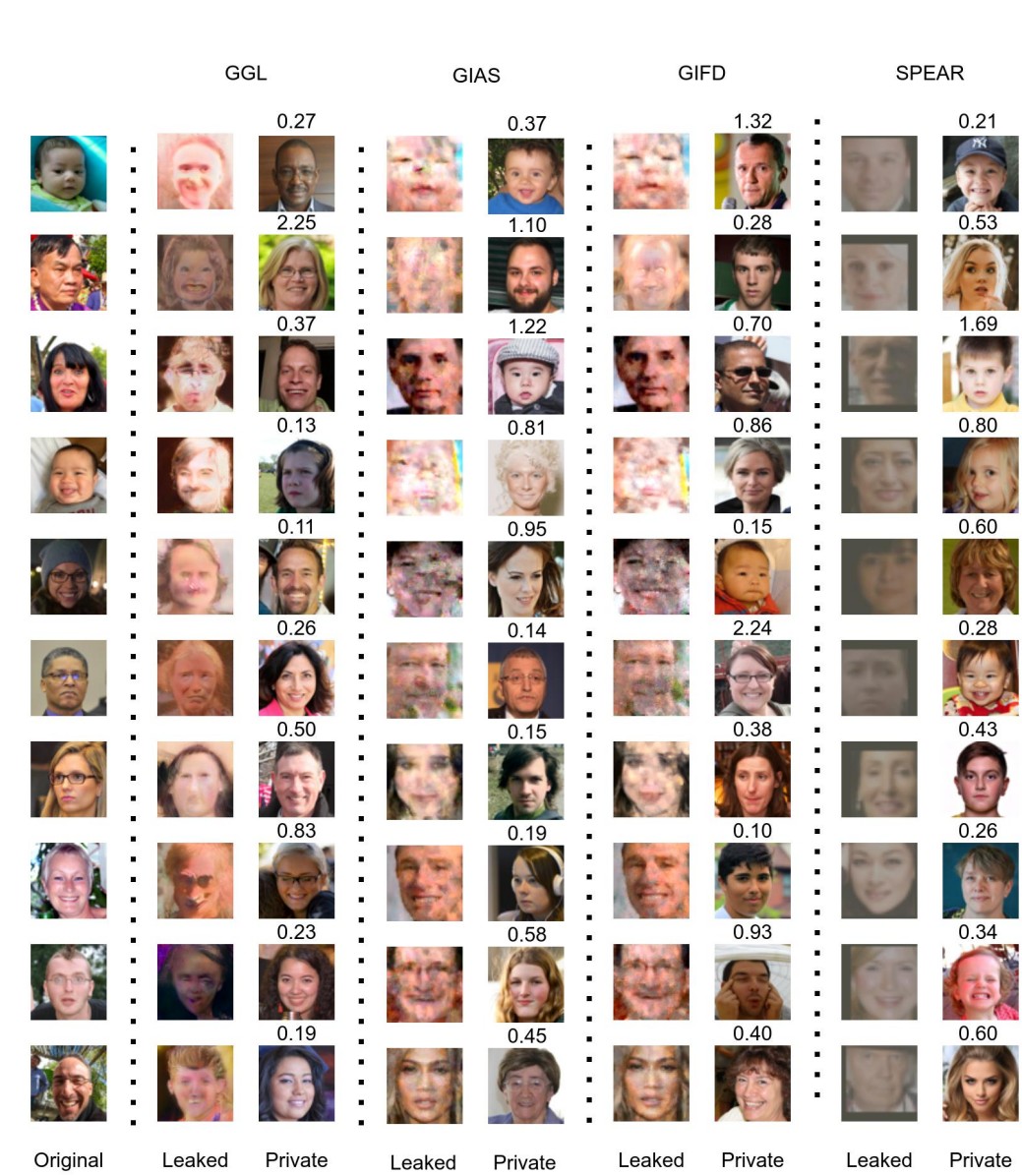

Figure 11: Reconstructed (leaked) FFHQ images under GradientHide using GIAs. The L2 distance between each reconstruction and its nearest retrieved private image is shown above the retrieved image.

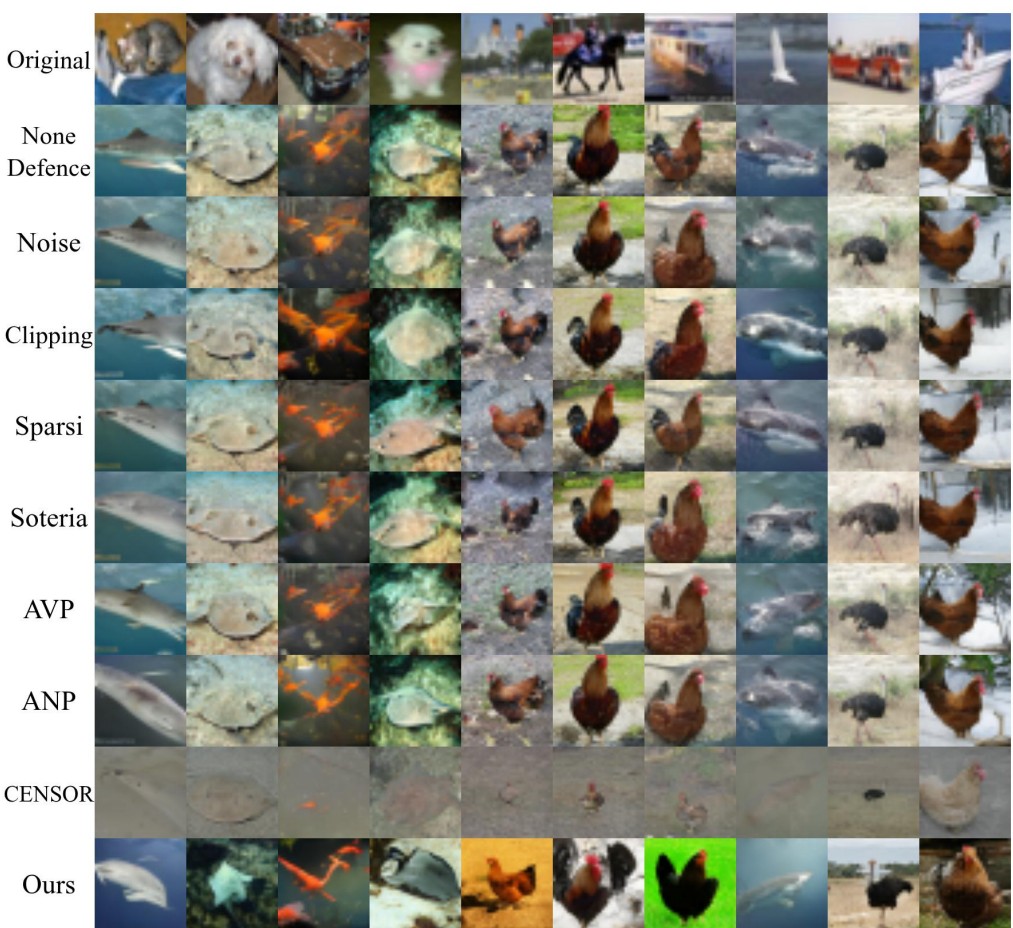

Figure 12: Visualizations of the Reconstructed Images from GGL on CIFAR-10.

1512
1513
1514
1515
1516
1517
1518
1519
1520
1521
1522
1523
1524
1525
1526
1527
1528
1529
1530
1531
1532
1533
1534
1535
1536
1537
1538
1539
1540
1541
1542
1543
1544
1545
1546
1547
1548
1549
1550
1551
1552
1553

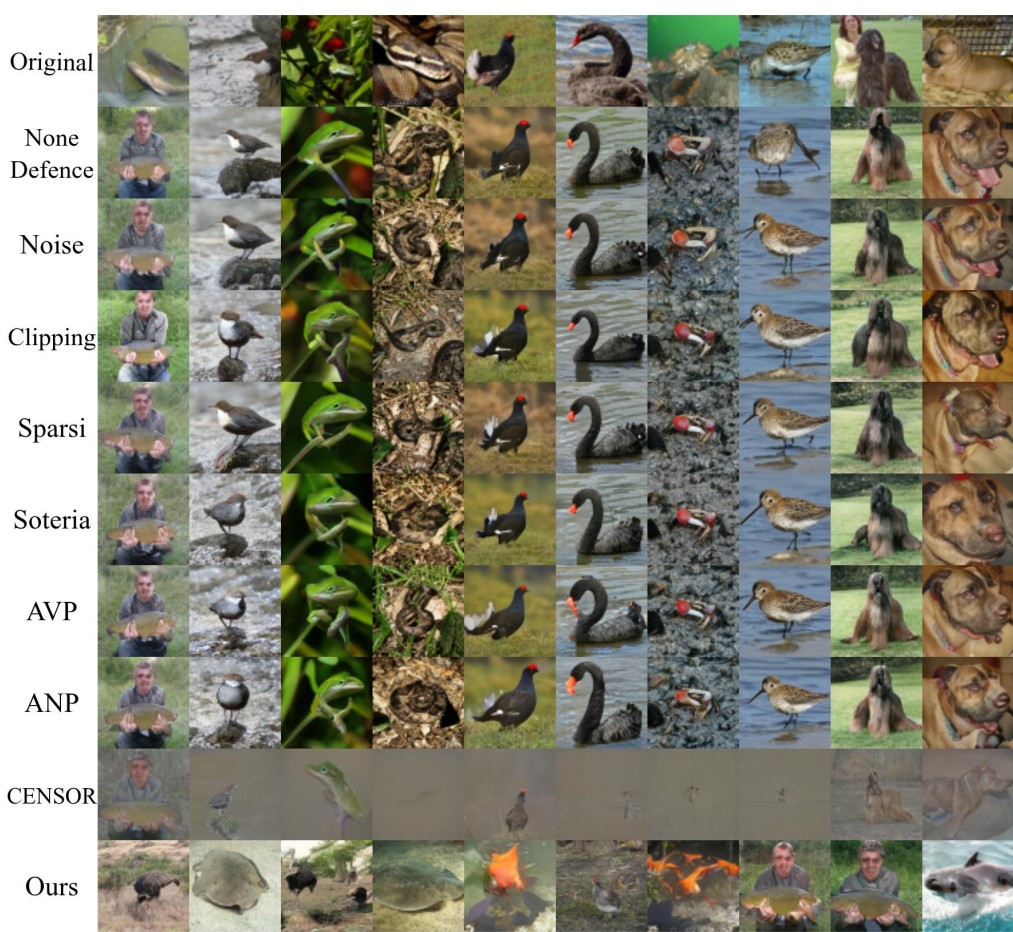

Figure 13: Visualizations of the Reconstructed Images from GGL on ImageNet.

1554
1555
1556
1557
1558
1559
1560
1561
1562
1563
1564
1565

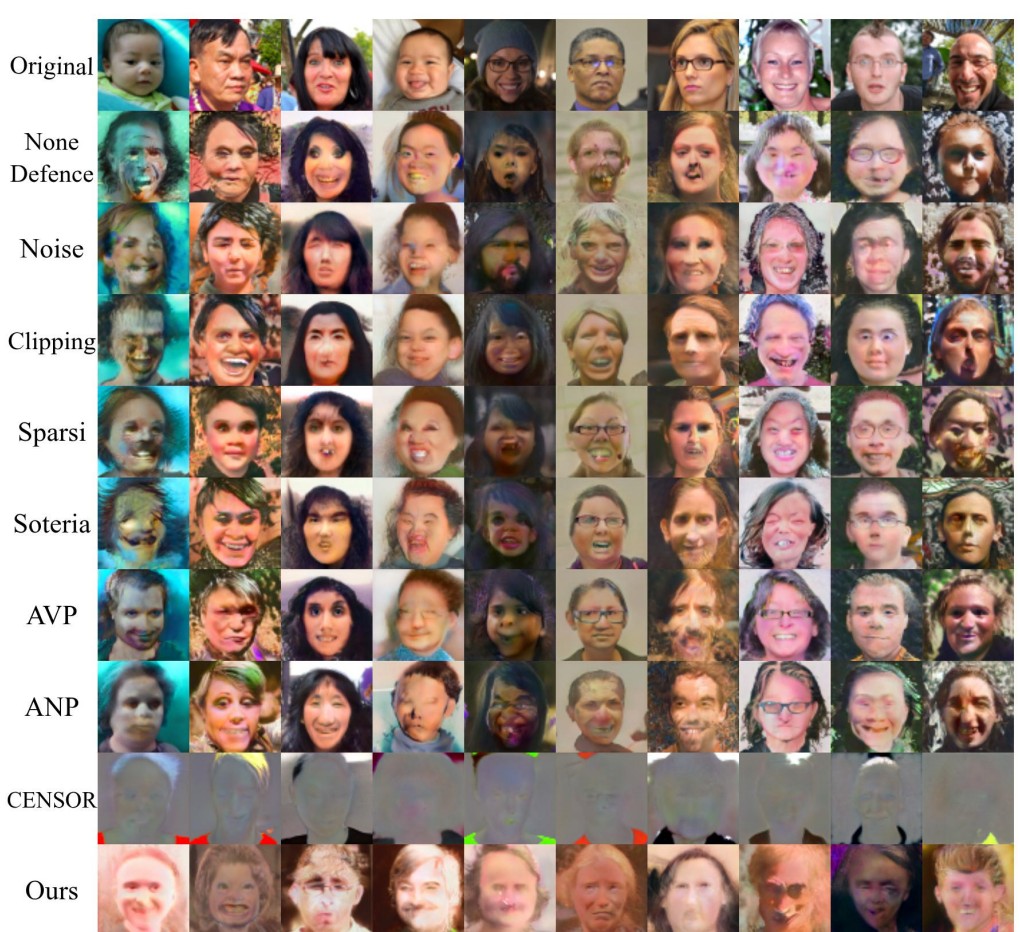

Figure 14: Visualizations of the Reconstructed Images from GGL on FFHQ.

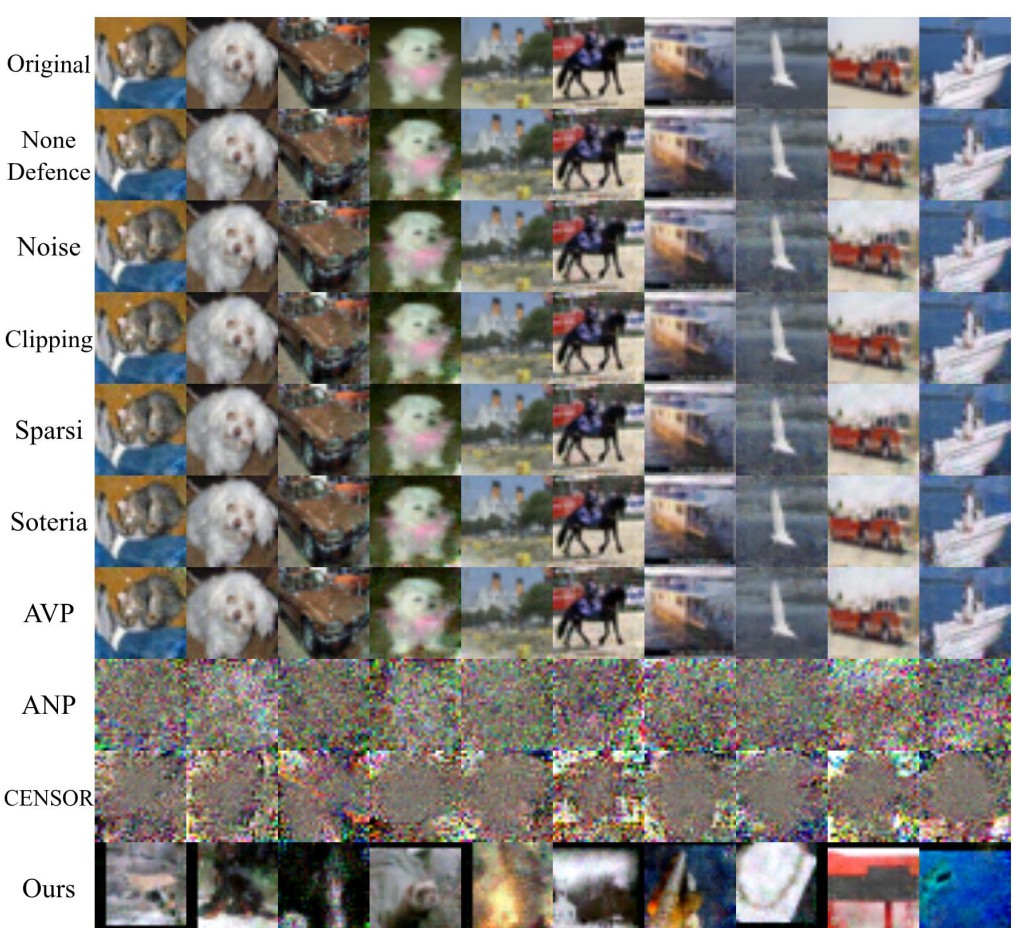

Figure 15: Visualizations of the Reconstructed Images from GIAS on CIFAR-10.

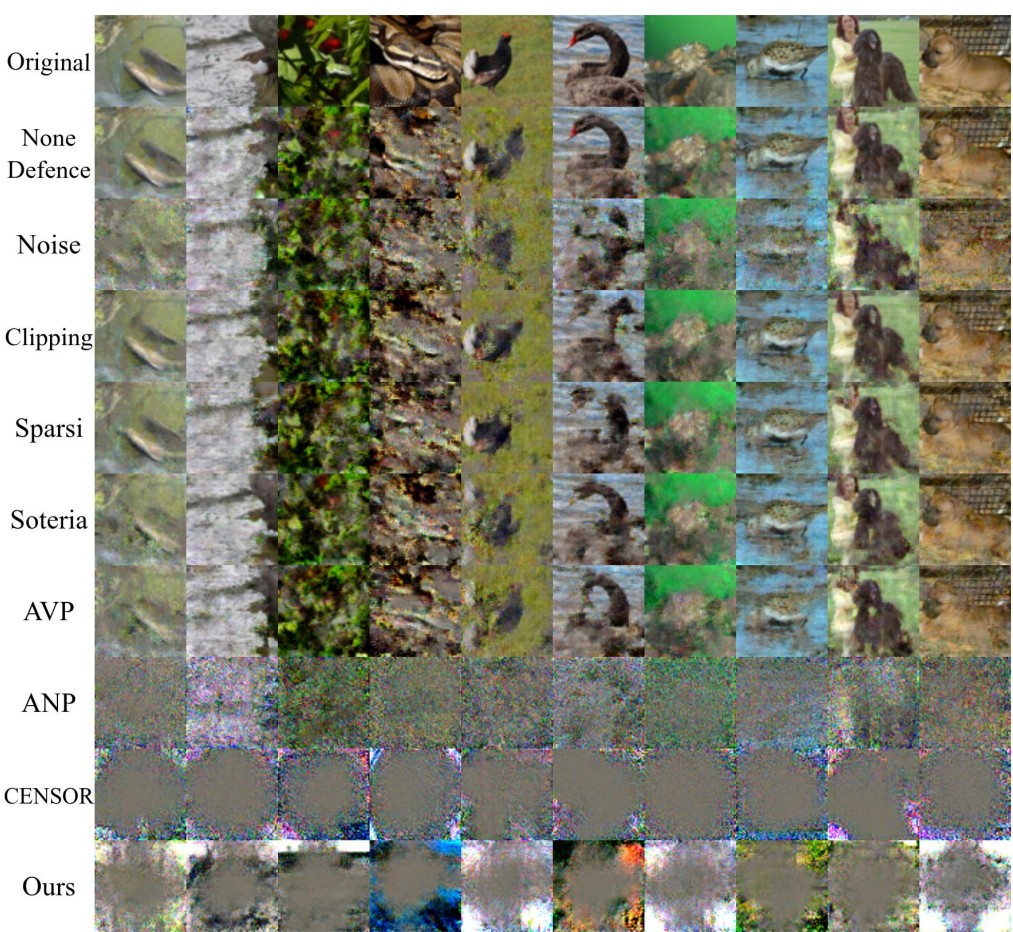

Figure 16: Visualizations of the Reconstructed Images from GIAS on ImageNet.

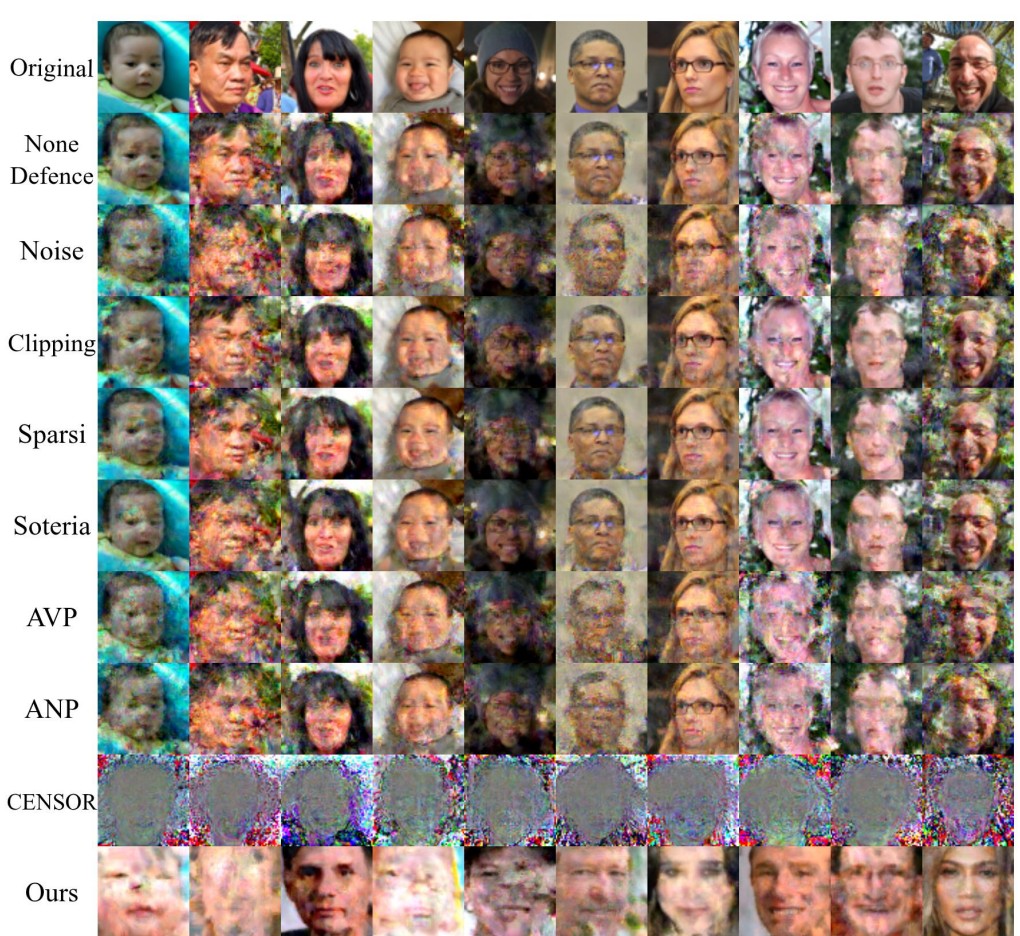

Figure 17: Visualizations of the Reconstructed Images from GIAS on FFHQ.

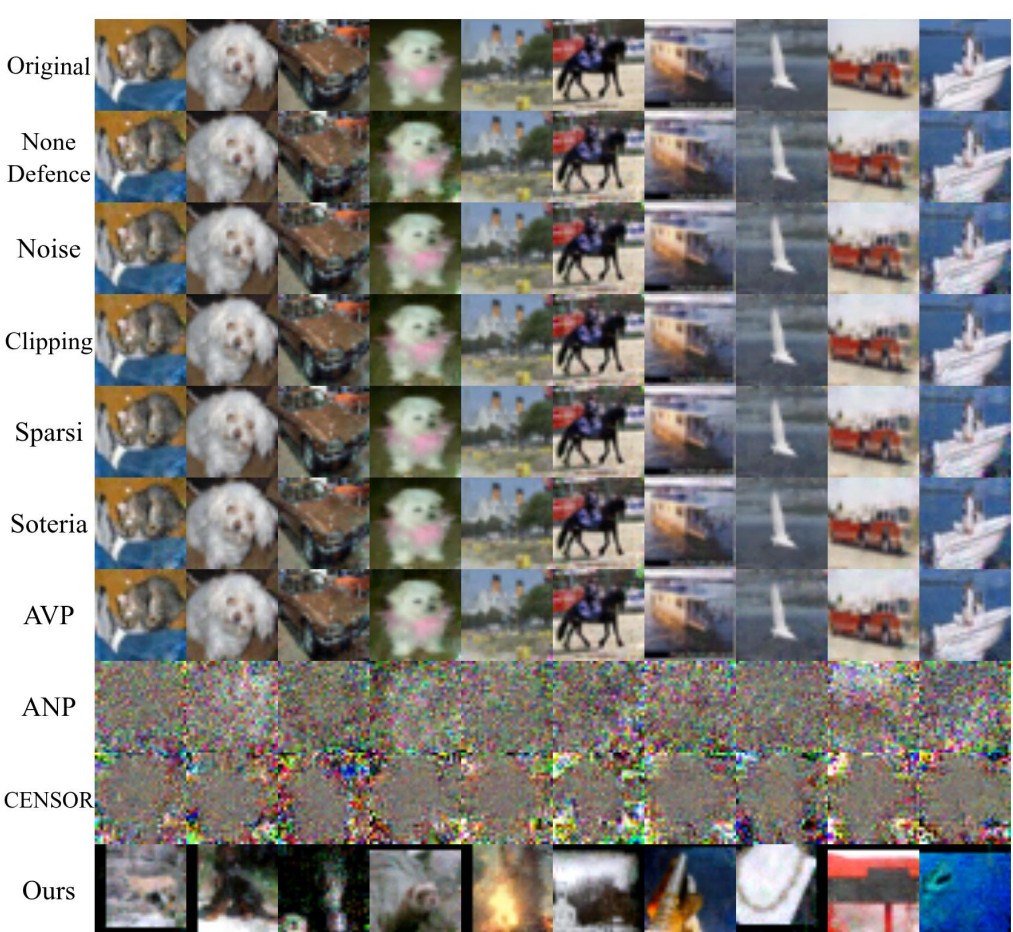

Figure 18: Visualizations of the Reconstructed Images from GIFD on CIFAR-10.

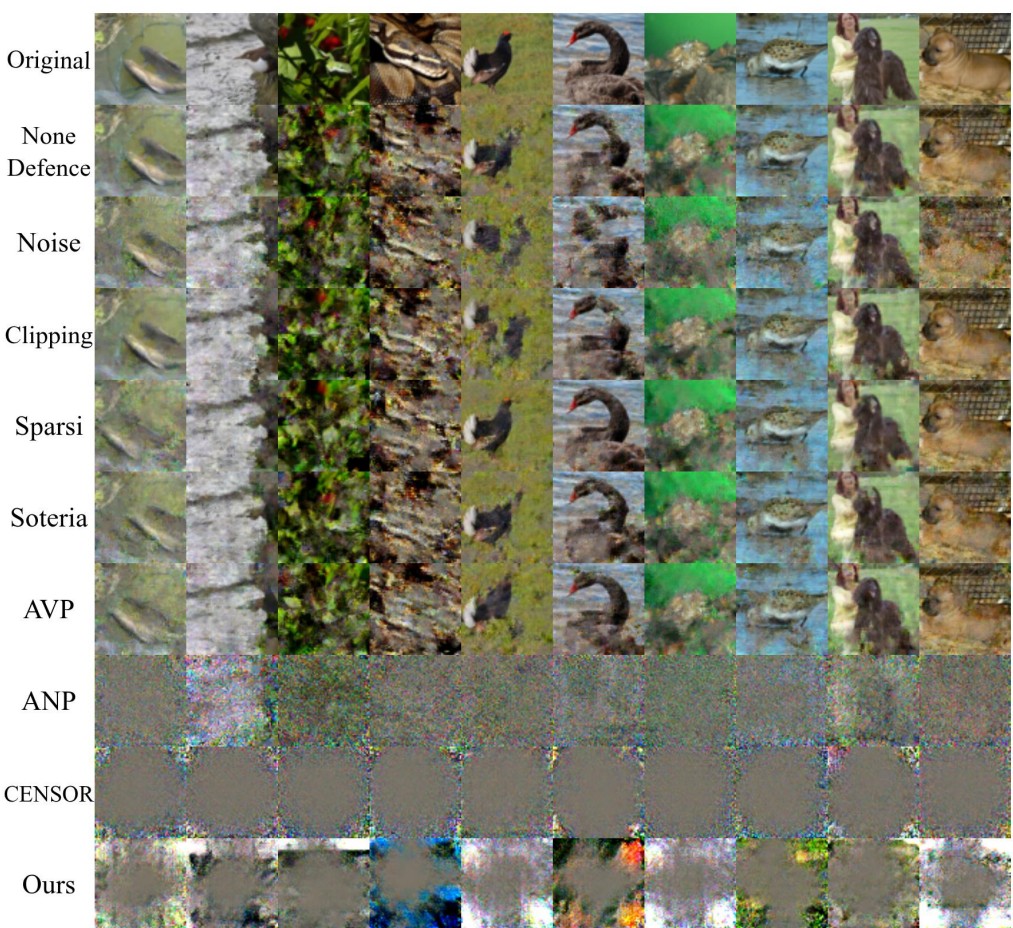

Figure 19: Visualizations of the Reconstructed Images from GIFD on ImageNet.

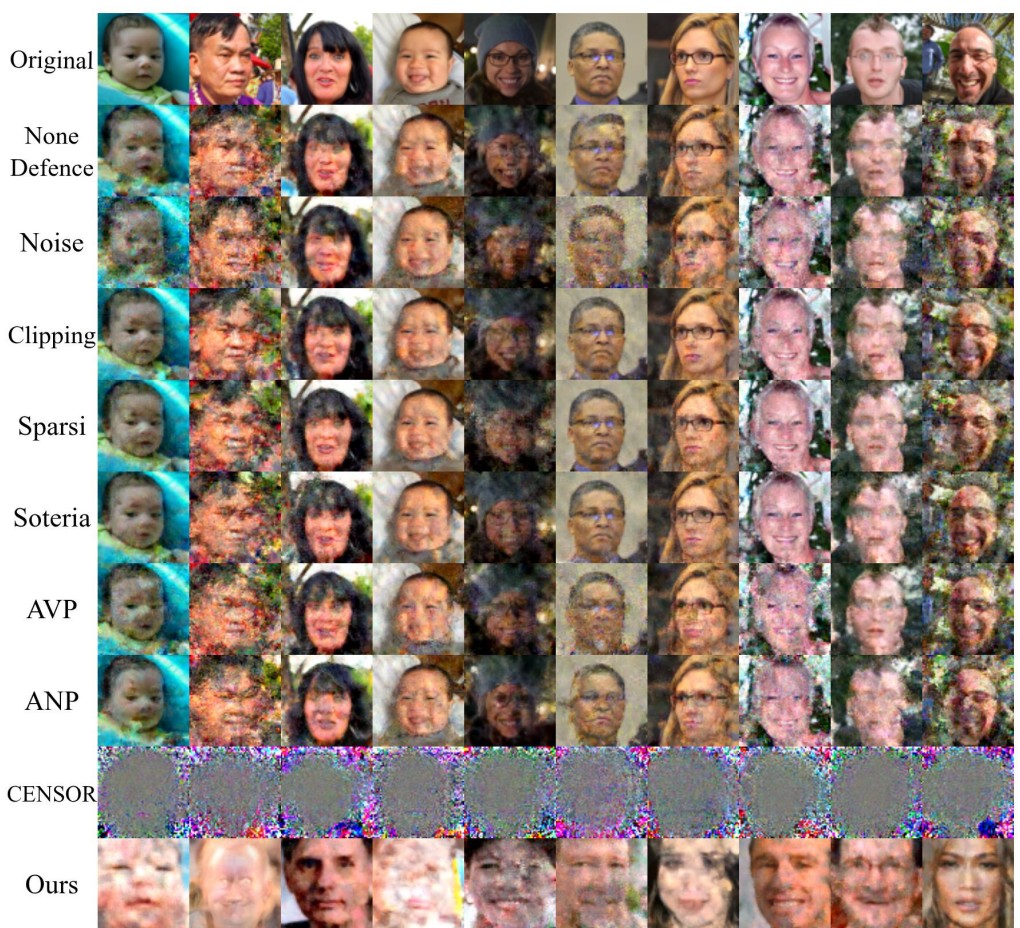

Figure 20: Visualizations of the Reconstructed Images from GIFD on FFHQ.

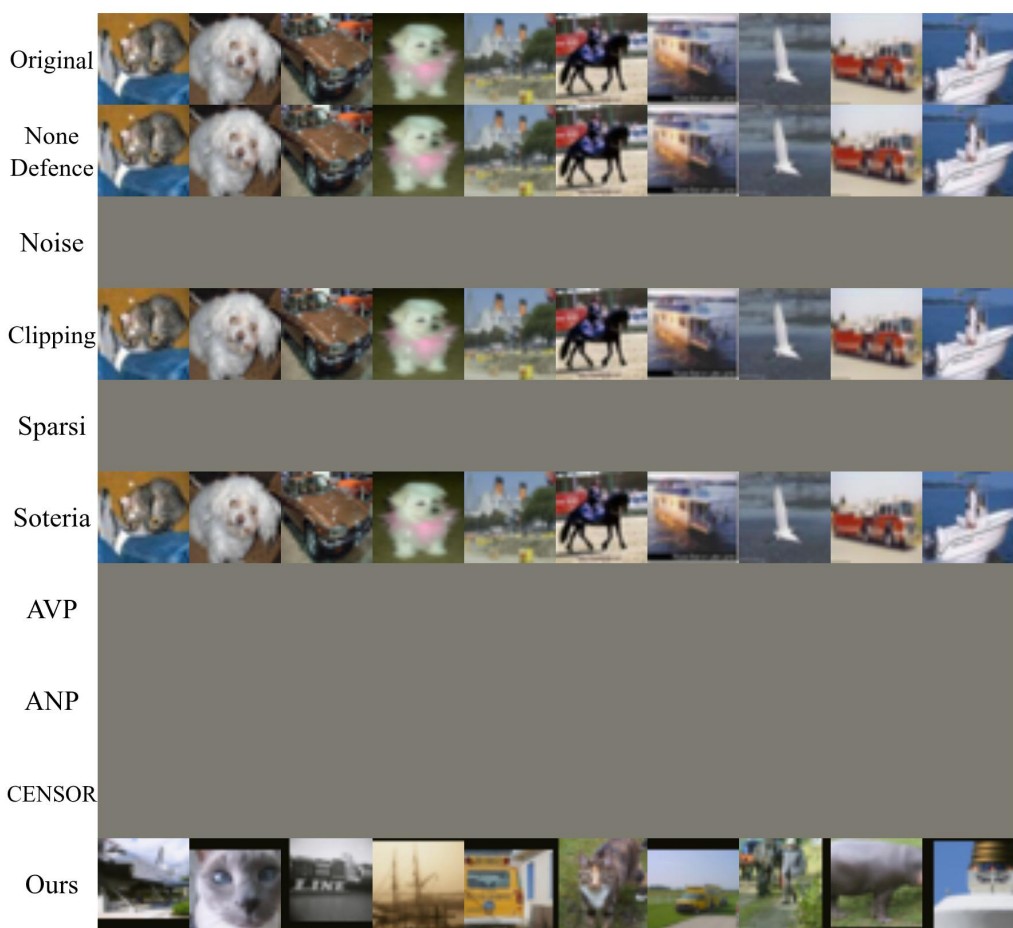

Original

None
Defence

Noise

Clipping

Sparsi

Soteria

AVP

ANP

CENSOR

Ours

Figure 21: Visualizations of the Reconstructed Images from SPEAR on CIFAR-10.

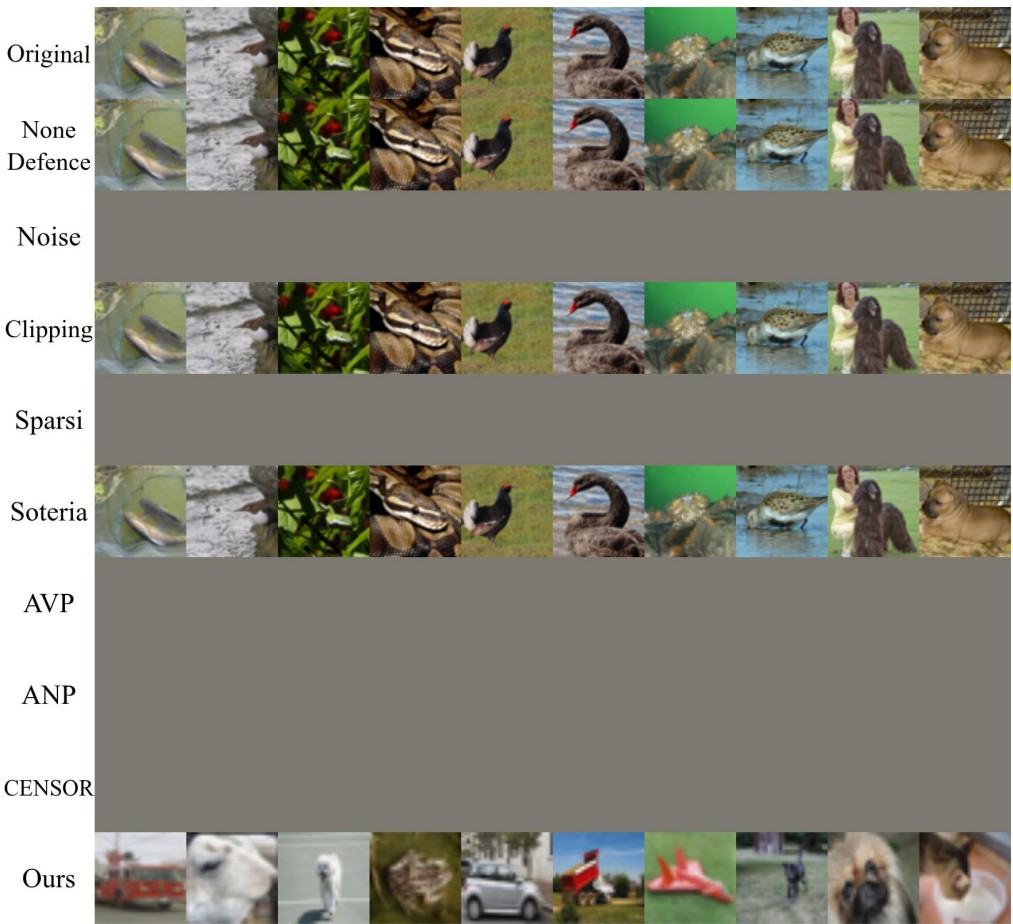

Figure 22: Visualizations of the Reconstructed Images from SPEAR on ImageNet.

Original

None
Defence

Noise

Clipping

Sparsi

Soteria

AVP

ANP

CENSOR

Ours

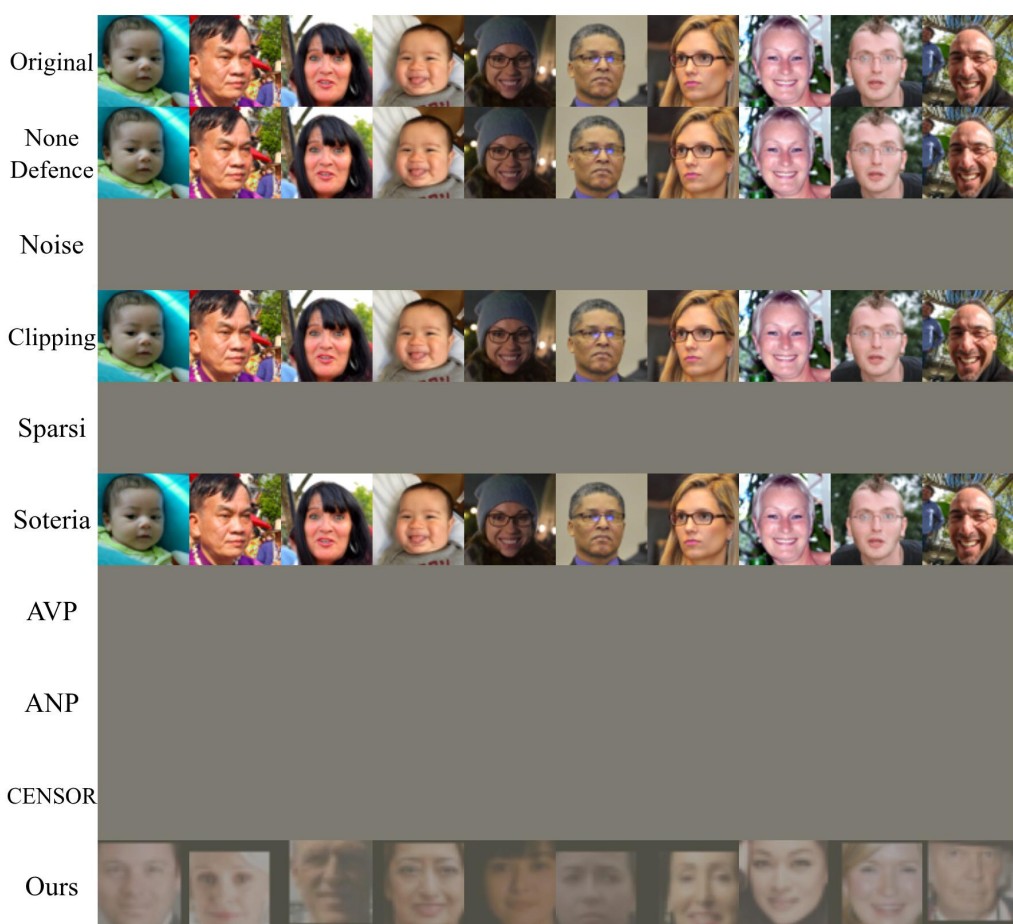

Figure 23: Visualizations of the Reconstructed Images from SPEAR on FFHQ.

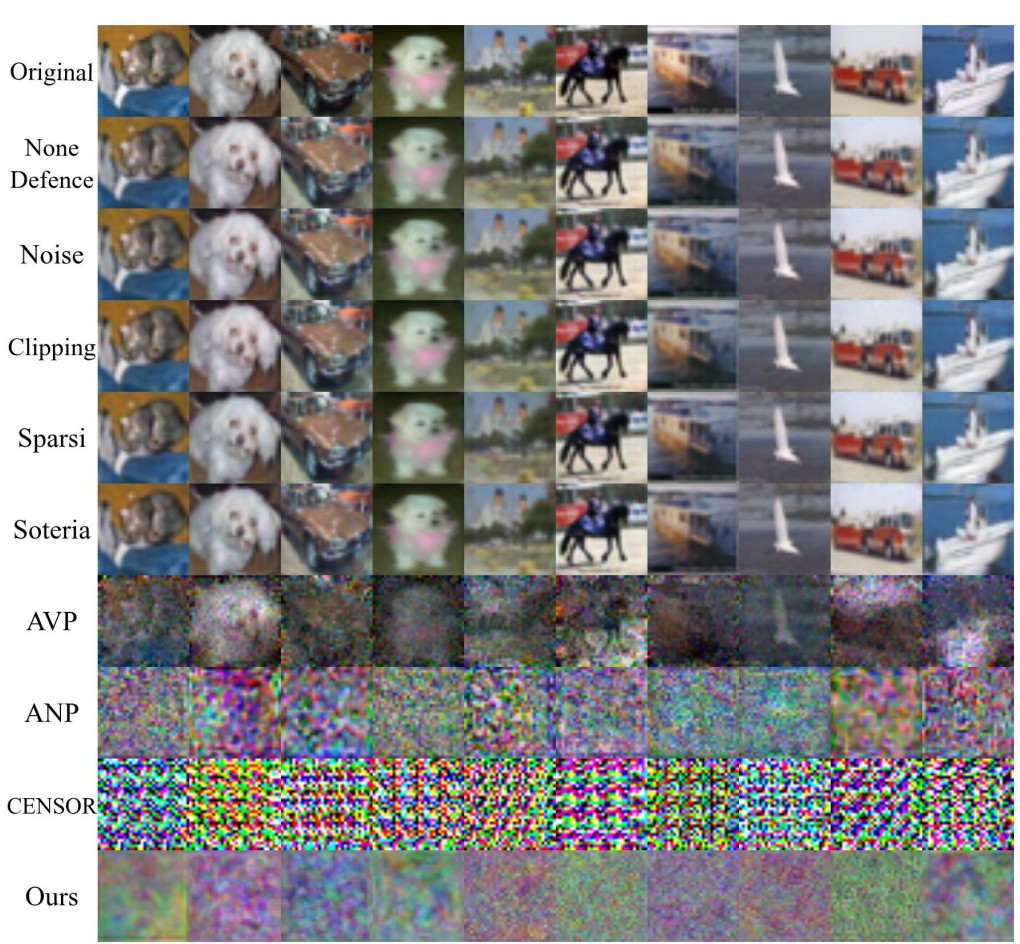

Figure 24: Visualizations of the Reconstructed Images from GI-NAS on CIFAR-10.

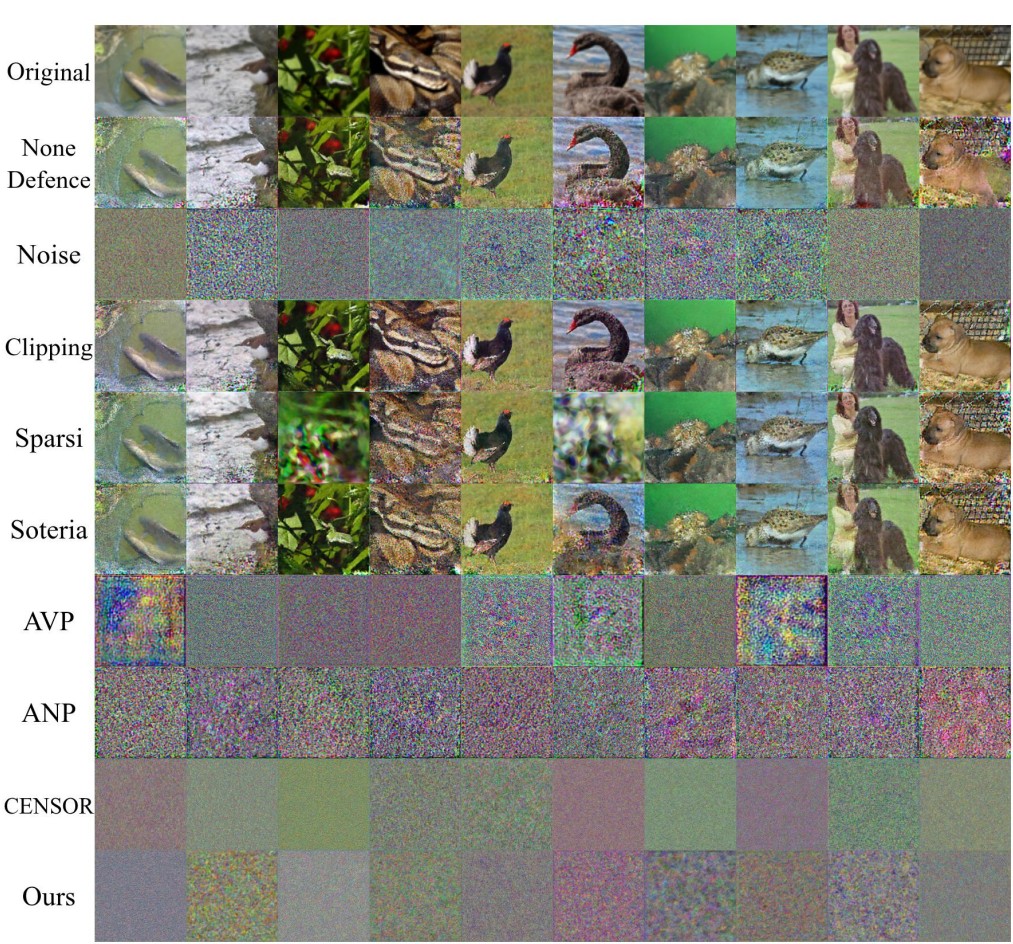

Figure 25: Visualizations of the Reconstructed Images from GI-NAS on ImageNet.

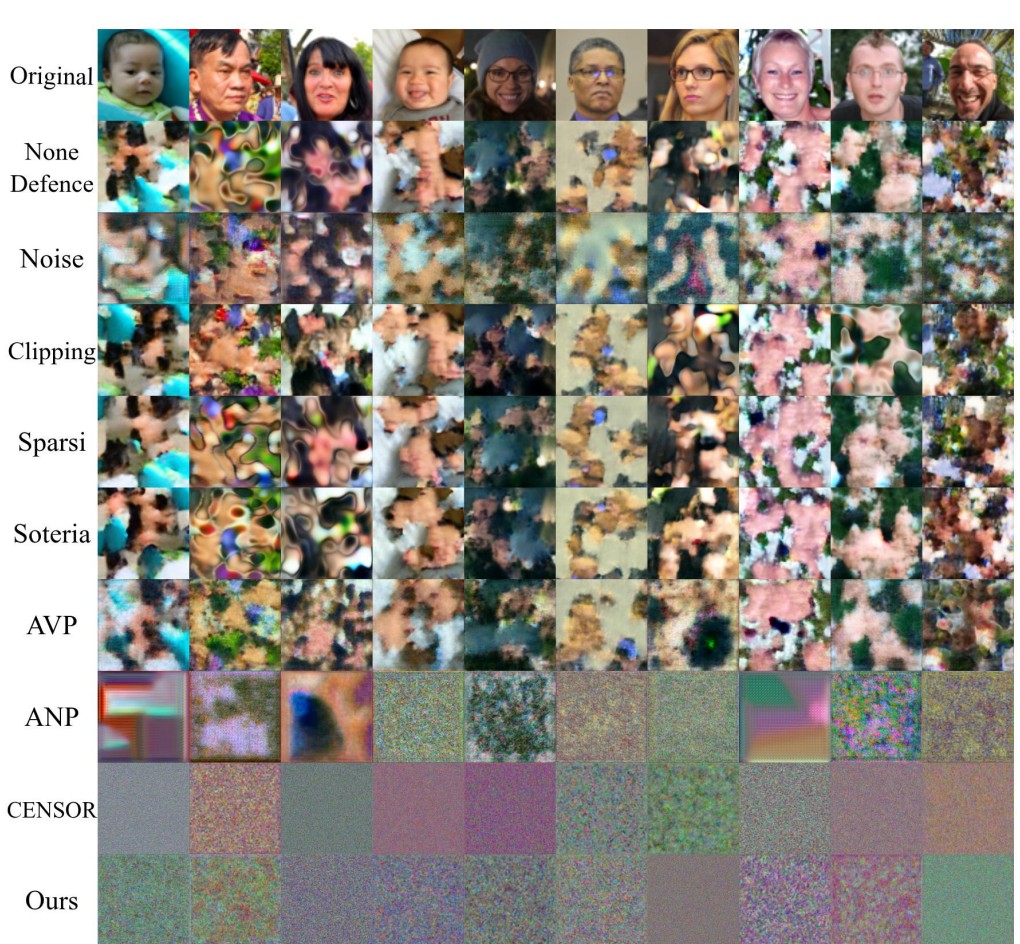

Figure 26: Visualizations of the Reconstructed Images from GI-NAS on FFHQ.

