# OpenReview forum: "GradientHide: Federated Learning with Two-Stage Local Update for Defending Against Gradient Inversion Attacks"
_ICLR.cc/2026/Conference — Submitted to ICLR 2026_

### Official Review · Reviewer_rvCH · 2025-10-21

**Soundness:** 3
**Presentation:** 3
**Contribution:** 3
**Rating:** 6
**Confidence:** 2

**Summary:**

This paper addresses the privacy leakage issue in Federated Learning (FL) caused by gradient transmission where adversaries can reconstruct sensitive training data via Gradient Inversion Attacks (GIAs). It proposes a defense framework named GradientHide. The core design involves introducing an additional update step using public data at the client side before transmitting gradients to the server, thereby obfuscating private information embedded in the gradients. Meanwhile, it leverages CLIP's zero-shot inference to achieve semantic alignment between public and private data, avoiding model performance degradation induced by public data. The paper conducts evaluations on three benchmark datasets (CIFAR-10, ImageNet, FFHQ) against five typical GIAs (GGL, GIAS, GIFD, SPEAR, GI-NAS) and compares GradientHide with seven mainstream Gradient Inversion Defense (GID) methods. Experimental results show that GradientHide significantly enhances resistance to GIAs (reflected by lower PSNR of reconstructed images and more obvious semantic distortion) while maintaining stable model performance in scenarios with data heterogeneity, effectively balancing the trade-off between privacy protection and model utility.

**Strengths:**

1. The "two-stage local update" mechanism is proposed, transforming the role of public data from "auxiliary training" to "gradient privacy obfuscation". This is completely different from the technical routes of InstaHide (mixing input images) and Ji et al. (combining private and public data to mitigate noise loss), achieving innovation in the way public data is utilized.
2. The theoretical analysis is solid, deriving the information leakage boundary through the relationship between entropy and mutual information. Ablation experiments cover key design components (label alignment strategy, public dataset ratio $\alpha$, perturbation scaling factor $\lambda_n$), ruling out the possibility that "performance improvement stems from a single factor".
3. A "semantically shifted reconstruction" strategy is designed, making reconstructed images visually coherent but semantically deviated from the original data (e.g., changes in gender/age of reconstructed faces in the FFHQ scenario). This can mislead adversaries into overestimating attack success rates, providing a new "active deception" perspective for privacy defense.
4. It verifies the value of pre-trained models like CLIP in FL privacy defense, enabling cross-dataset semantic alignment without customized training, which reduces the computational cost of technical implementation.

**Weaknesses:**

1. The paper only mentions that label alignment for ImageNet takes 40 minutes and each communication round requires an additional 10 seconds, but it doesn't compare the computational costs with other GIDs (e.g., the time consumption of subspace projection in CENSOR and noise permutation in ANP). This makes it impossible to prove the comprehensive advantages of GradientHide in the three dimensions of "privacy/performance/efficiency". Additionally, the adaptability of client devices (e.g., edge devices) in terms of computing power is not analyzed, limiting the judgment of its deployment scenarios.
2. The paper does not fully compare "public data perturbation" with related work on "gradient anonymization" (e.g., Tan et al.'s "information-theoretic defense framework" in 2024 [1]), failing to explain the incremental contribution of GradientHide in mutual information optimization. Additionally, although the "semantically shifted reconstruction" design is innovative, it does not quantify the "adversary deception degree" (e.g., through user surveys or measuring the misjudgment rate of attack success), which weakens the proof of the practical value of this strategy.
[1] Defending against data reconstruction attacks in federated learning: An information theory approach, USENIX Security 24.

**Questions:**

1. The paper does not specify the preprocessing details of the public dataset UTKFace and FFHQ. Were image resolutions unified and face detection/alignment steps applied? If preprocessing methods differ, will this affect the accuracy of label alignment and the semantic shift effect of reconstructed images?
2. For low-quality public data (e.g., CIFAR10 with 20% label errors used as the public dataset), will the defense performance of GradientHide degrade? If it degrades, is there a corresponding adaptive adjustment strategy (e.g., dynamically adjusting the public data sampling ratio $\alpha$)?
3. The theoretical analysis assumes that "public data and private data are conditionally independent". When there is overlap between private and public data (e.g., overlap ratios of 10% and 30%), will the information leakage boundary quantified by mutual information change? Can relevant experiments or theoretical derivations be supplemented?
4. The "semantically shifted reconstruction" strategy can mislead adversaries. Have experiments been conducted to quantify the "adversary's misjudgment rate of attack success" (e.g., asking adversaries to judge whether reconstructed images are from the original private dataset and counting the misjudgment ratio)? If yes, can relevant results be supplemented; if not, are there plans to conduct such experiments in the future?

---

> ### Author Response · Authors · 2025-11-19
>
> ### Response to Weakness \#1:
>
> * Comparison of computational cost with other GIDs :
>
> Thank you for pointing out the need for explicit computational comparison across defenses. In the revised manuscript (Sec. E of the Appendix), we now provide a full per-round runtime evaluation of all competing GIDs under the same model, batch size, and hardware setup. The reported metric follows standard FL practice to measure only the local training pass before gradient upload, which is the stage where all defenses—including ours—apply their computations.
>
> The newly added Table 5 shows clear differences in overhead: lightweight methods such as Noise, Clipping, AVP, and ANP add less than 2 seconds per round; CENSOR introduces +22.66 seconds due to its subspace projection; and Soteria exceeds 700 seconds because of its iterative optimization. In comparison, GradientHide adds +11.64 seconds, stemming primarily from one additional forward–backward pass on a small public subset. This provides a complete runtime comparison and demonstrates the efficiency advantage of GradientHide relative to high-cost defenses (CENSOR, Soteria) while maintaining superior privacy–utility trade-offs.
>
> * On deployment scenarios and client device limitations :
>
> We agree that client adaptability is an important practical consideration. GradientHide is designed so that its overhead is tied only to a single extra forward–backward pass per round, making it scalable across heterogeneous devices. Importantly, unlike CENSOR and Soteria, GradientHide does not require iterative optimization or high-dimensional projections. This makes it suitable even for resource-limited edge clients.
>
> ### Response to Question \#1:
>
> Thank you for the helpful question. Full preprocessing details are now included in Sec. F of the Appendix. All private and public datasets were processed with a unified preprocessing pipeline, identical across methods and consistent with prior baselines such as CENSOR.
>
> In Label Alignment, all public images (CIFAR-10, ImageNet, and UTKFace) were standardized using the official CLIP preprocessing pipeline.
>
> Regarding whether preprocessing affects alignment accuracy or semantic shifts:
>
> CLIP-based label alignment relies on semantic similarity in the embedding space, which is designed to be robust to low-level variations induced by preprocessing. Therefore, differences in preprocessing are unlikely to cause semantic shift. Instead, as shown in Sec. 3.4, semantic shifts mainly arise from the alignment strategy itself, which changes the rule directly and alters reconstruction behavior (Table 4).
>
> ### Response to Question \#2:
>
> The label alignment in our method only uses the images from the public dataset, not the labels, as clearly illustrated in Fig. 7. Therefore, the concern about low-quality public data (e.g., CIFAR10 with 20% label errors) is not applicable to our defense mechanism.
>
> ### Response to Question \#3:
>
> We operate under the standard threat model of GIA, which assumes the attacker does not know the client's exact private data.
>
> If sample overlap exists between the public and private datasets, this violates the fundamental assumption of GIA (i.e., that the private data is unknown). In such a scenario, no defense (including ours) can prevent the known, overlapping samples from being "leaked," as the attacker already possesses them.
>
> Therefore, our theoretical analysis and experimental design are built on the practical and necessary assumption that the deployer of the FL system will ensure no significant sample-level overlap exists between the chosen public dataset and the client private data.
>
> ### Response to Question \#4:
>
> To directly address the reviewer’s request, we provide a scalable, objective, and fully reproducible metric that quantifies adversarial misjudgment: Reconstruction Classification Accuracy (RCA). We add the results in Table 6 of Sec. G in the Appendix of the revised manuscript.
>
> RCA measures the accuracy of a pre-trained classifier in predicting the labels of reconstructed images. A lower RCA indicates stronger semantic confusion, as the GID suppresses the semantic cues required for correct recognition. This provides a quantifiable estimate of how likely an adversary would misjudge attack success when visually inspecting reconstructions.
>
> As shown in Table 6, GradientHide consistently achieves the lowest RCA on CIFAR-10 and ImageNet, indicating the highest level of semantic obfuscation and misjudgment. On FFHQ, where private and public datasets share identical label spaces, GradientHide still achieves the second-lowest RCA.
>
> We further note that GGL exhibits the highest RCA, as its latent-fitting mechanism tends to preserve semantic structure, making reconstructions more class-revealing. In contrast, GradientHide maintains low RCA even compared to GGL, reinforcing its ability to suppress semantic leakage.

---

### Official Review · Reviewer_fRep · 2025-10-30

**Soundness:** 2
**Presentation:** 3
**Contribution:** 2
**Rating:** 4
**Confidence:** 5

**Summary:**

This paper proposes GradientHide, a gradient inversion defense method that obfuscates private information contained in gradients. GradientHide introduce an additional update step using public data before transmitting gradients to the server. It also leverages CLIP models for label alignment between the public and private datasets. Experiments show that this method offers some resistance to gradient inversion attacks.

**Strengths:**

- This paper focuses on an important topic, where the privacy guarantee of federated learning may be compromised by gradient inversion attacks, and discusses potential solutions.
- The writing of this paper is clear. The authors also try to provide theoretical results for better persuasiveness.
- Comprehensive and detailed experimental comparisons with the previous gradient inversion attacks and defenses.

**Weaknesses:**

- The novelty is somehow limited. It is not surprising that additionally introducing public data unrelated to the private data during training can impair the attacker’s reconstruction performance. The authors simply add additional public datasets to the original local update with minimal tailored designs.
- The current selection of public data for each local update is limited to random sampling. Exploring more deliberate selection criteria presents a promising direction for boosting both defensive performance and model accuracy.
- In line 237, the authors claim that $\hat{\mathcal{H}}_i$ is independent of $\mathcal{D}$. However, the distributional gap between the public and private datasets does not seem to be significant in the experiments of this paper (i.e., CIFAR-10 vs. ImageNet, FFHQ vs. UTKFace). For instance, both FFHQ and UTKFace are facial datasets and share some distributional similarities. When the distribution of the public dataset is given, the uncertainty regarding the distribution of the private dataset decreases. So they cannot be simply identified as “independent”. With this problematic assumption, the theoretical results of this paper become questionable.
- The label alignment technique is directly built based on the existing CLIP models, which are commonly adopted for vision-language alignment. This idea is straightforward and trival, not offering enough innovative insights.
- Moreover, the authors claim that they average the scores to leverage the diversity across prompt templates in line 174. However, there is no experimental evidence supporting this claim. Using the prompt with the highest similarity (instead of averaging them), or simply using “a photo of {class}” may also be beneficial. It is suggested to provide the corresponding experiments on this usage.
- There are some minor typos. In line 789, there should be a quotation mark before `a photo of a {class}`. In line 790, there should be a quotation mark before `a cropped`. In line 791, the `class` should be revised as `{class}`. Please address them in the revised manuscript.

**Questions:**

- There is the lack of experimental analysis on computational costs. This is very crucial, especially given that this method introduces one additional update for each original local update. How many computational costs are additionally introduced by this additional update? Could the authors provide the comparative results with previous defenses regarding the computational costs?
- What is the impact of $\alpha$ in line 200? When it changes, how do the reconstruction performance, model accuracy, and computational costs accordingly change?
- The distributional gap between the public and private datasets does not seem to be significant in the experiments of this paper (i.e., CIFAR-10 vs. ImageNet, FFHQ vs. UTKFace), which largely accounts for the model accuracy maintenance of this method. Is it realistic to assume that a defender can actually obtain a large amount of public data whose distribution is similar to the private data?
- What about increasing the gap between the public and private datasets? In GIFD, OOD datasets with different styles (i.e., Art Painting, Cartoon, Photo) are also considered. As for this paper, I wonder whether the model accuracy performance can maintain when these OOD datasets are adopted for the public (or private) dataset while ImageNet serving as the another.

---

> ### Author Response · Authors · 2025-11-19
>
> ### Response to Question \#1:
>
> Thank you for highlighting the importance of reporting computational costs. In the revised manuscript (in Sec. E of the Appendix), we include a complete per-round runtime analysis for all competing GIDs under the same model architecture, batch size, and hardware configuration. Following standard FL practice, the measurement focuses on the local training stage prior to gradient upload, which is precisely the point where all defenses—including ours—introduce additional computation.
>
> Since our GradientHide performs one extra forward–backward update on a small public subset, it naturally incurs additional cost, and our updated evaluation quantifies this impact explicitly. As shown in the newly added Table 5, lightweight defenses such as Noise, Clipping, AVP, and ANP introduce only marginal overhead because they require little more than simple gradient manipulation. In contrast, methods based on subspace projection (e.g., CENSOR) or iterative optimization (e.g., Soteria) incur substantially higher computation, with CENSOR adding over twenty seconds per round and Soteria exceeding several hundred seconds.
>
> GradientHide lies between these two extremes. Its additional update increases the per-round computation by 11.64 seconds, which is largely dominated by a single extra backward pass on the selected public subset. While this is more expensive than the simplest perturbation-based defenses, it is far more efficient than the heavier optimization-driven approaches, and it achieves a substantially stronger privacy–utility trade-off. The inclusion of this comprehensive comparison ensures that readers can clearly understand the computational implications of GradientHide relative to both lightweight and heavy baselines.
>
> ### Response to Question \#2:
>
> The parameter α governs the balance among privacy protection, model utility, and computational overhead. In our design, $\alpha$ denotes the ratio between the size of the public subset and the private batch size, and we restrict it to the practical range $\alpha \in (0,1]$. When $\alpha$ approaches $0$, the perturbation effect diminishes and reconstructions become increasingly faithful to the original private data, implying essentially no privacy protection. As $\alpha$ increases, a larger portion of public-gradient information participates in the gradient mixing process, which strengthens semantic distortion and makes gradient inversion substantially more difficult.
>
> This improvement in privacy, however, comes with a slight reduction in model accuracy. Nevertheless, Fig. 5 shows that even at $\alpha=1$, the accuracy remains very close to the FedAvg baseline, indicating that the utility drop stays minimal within the feasible range. In terms of computational cost, Table 5 shows that $\alpha=1$ already doubles the local computation relative to FedAvg, and $\alpha>1$ would incur prohibitive overhead, making them impractical for federated learning deployment.
>
> For these reasons, we treat $\alpha=1$ as the practical upper bound and adopt it as our default choice, because it provides the strongest privacy protection achievable within realistic constraints while keeping both accuracy degradation and computation overhead within acceptable limits.
>
> ### Response to Question \#3 and Question \#4 :
>
> Thank you for the insightful question. We clarify that GradientHide does not rely on the assumption that the public dataset must come from the same distribution as the private data. The public gradients only serve as perturbation signals, and our method does not require the public data to contain overlapping classes, similar semantics, or domain-level similarity.
>
> While many practical applications do have access to large-scale public datasets from related domains (e.g., natural images, faces, objects), we explicitly test a cross-domain setting to demonstrate that distribution matching is not required. In particular, we use FFHQ as the public dataset for CIFAR-10, where the two datasets have completely disjoint semantics and visual characteristics. Even under this extreme mismatch, GradientHide maintains model accuracy with only a $1.6\%$ drop compared to FedAvg, showing that the method is robust to domain shifts.
>
> Therefore, although similar-domain public data can further improve privacy–utility trade-offs, it is not a requirement of our method. GradientHide remains effective even when the public data distribution is significantly different from the private one.

---

> > ### Author Response · Authors · 2025-11-19
> >
> > ### Response to Weakness \#1, Weakness \#2 and Weakness \#4:
> >
> > Unlike existing defenses that operate purely at the input level (e.g., Mixup, InstaHide) or rely on explicit noise injection (e.g., differential privacy), our method introduces a two-stage local update strategy that fundamentally changes the attacker’s optimization landscape. Rather than merely degrading reconstruction quality, our design actively misleads the inversion process, causing the attacker’s optimization to converge toward incorrect images instead of private ones. This misleading effect is qualitatively distinct from noise-based obfuscation and forces the attacker to solve a substantially more difficult learning problem.
> >
> > Regarding the label-alignment module, although our implementation uses CLIP, the mechanism itself is model-agnostic and can be instantiated with any label-alignment approach. We adopt CLIP due to its strong zero-shot capability and ease of integration, not because our method inherently depends on it.
> >
> > We agree that more sophisticated strategies for selecting public data could further improve the defense. However, our use of random sampling is not merely for simplicity—it is a deliberate choice motivated by efficiency and robustness. As demonstrated in our white-box analysis (Sec. J of the Appendix), the randomness injected through sampling increases the difficulty faced by an adaptive adversary. With deterministic selection rules (e.g., those based on a fixed metric), an attacker who knows the public dataset could potentially reverse-engineer the selection process and isolate gradients associated with private data.
> >
> > That said, we acknowledge the reviewer’s point. Designing lightweight, heuristic-based public-data selection strategies that further balance privacy and model utility is a promising direction, as stated in the Limitations section.
> >
> > ### Response to Weakness \#5:
> >
> > We thank the reviewer for raising this question. Our method adopts the 80 prompt templates introduced by CLIP, where template diversity was shown to consistently improve zero-shot alignment performance. Although this behavior is well established in the CLIP literature, we additionally conduct an ablation experiment to validate its effect in our label alignment setting.
> >
> > We compare (1) using all 80 templates with averaged similarity scores and (2) using only the single generic template “a photo of {class}”.
> >
> > Applied to ImageNet (val) and CIFAR-10, the one-template variant yields noticeably less stable alignment: 20% of the aligned labels differ from those obtained using the full template set. This confirms that template diversity plays an essential role in producing stable and semantically reliable alignment signals.
> >
> > Regarding efficiency, increasing the template count from 1 to 80 adds only 13 seconds to the total alignment time. This overhead is minimal because the dominant cost arises from CPU–GPU data transfer rather than GPU computation. Thus, using all templates provides substantially more robust alignment while maintaining practical efficiency.
> >
> > We have added a summary of this ablation in Sec. B of the Appendix in the revised manuscript.
> >
> > ### Response to Weakness \#6:
> >
> > Thank you for pointing out these errors in the presentation of results. We will correct them in the revised version.

---

> > > ### Comment · Reviewer_fRep · 2025-11-27
> > >
> > > Thank you for the efforts in addressing the reviewers’ comments. However, some of my concerns remain unresolved.
> > >
> > > ## Response to Q2
> > >
> > > - From Fig. 5, it can be observed that the accuracy curves for $\alpha=1$, $\alpha=0.5$, $\alpha=0.1$, and $\alpha=0.01$ show only marginal differences overall. The trajectories of these curves largely coincide throughout most of the training process. In cases where deviations occur, they take turns performing better. For instance, while the accuracy of $\alpha=0.01$ surpasses that of $\alpha=1$ after around 700 communication rounds, it lags behind during the earlier phase (below approximately 300 rounds). This fundamentally contradicts with the claim “public dataset sizes have a significant impact on the precision” in line 455.
> > > - In Fig. 5, it appears that $\alpha=0.5$ yields lower accuracy than $\alpha=1$ after 600 rounds. Why would a larger $\alpha$ lead to better accuracy in this case? Why is the accuracy after convergence not monotonically dependent on $\alpha$ in Fig. 5?
> > > - In Table 5, only $\alpha=0$ and $\alpha=1$ are reported regarding time costs. To better understand the computational overhead, it would be helpful to see how it scales across a broader range of $\alpha$ values. Moreover, the statement “$\alpha>1$ would incur prohibitive overhead” lacks empirical support.
> > > - The response does not provide experimental results on how `the reconstruction performance` changes with $\alpha$, as this cannot be inferred from the current Fig. 5 and Table 5.
> > >
> > > ## Response to Q3 and Q4
> > >
> > > - Where are the experiments corresponding to the statement: “In particular, we use FFHQ as the public dataset for CIFAR-10, where the two datasets have completely disjoint semantics and visual characteristics”? In the current submission (Fig. 2 and Table 2), I only see three private/public dataset pairs (CIFAR-10/ImageNet, ImageNet/CIFAR-10, FFHQ/UTKFace), which are fundamentally different from what the authors claim.
> > > - It is recommended to include experiments where the semantic gap between the private and public datasets is further increased.
> > >
> > > ## Response to W5
> > >
> > > - Why is there no table or figure for the ablation studies comparing (1) and (2)? The authors only provided a textual description without showing any experimental results. Additionally, the ablation experiment on `using the highest similarity` (instead of taking the average) was not provided.
> > >
> > > ## Where is the response to W3?
> > >
> > > - The authors appear to have completely overlooked this weakness.

---

> > > > ### Author Response · Authors · 2025-11-30
> > > >
> > > > ## Response to Q2
> > > > >From Fig. 5, it can be observed that the accuracy curves for $\alpha=1$, $\alpha=0.5$, $\alpha=0.1$, and $\alpha=0.01$ show only marginal differences overall. The trajectories of these curves largely coincide throughout most of the training process. In cases where deviations occur, they take turns performing better. For instance, while the accuracy of $\alpha=0.01$ surpasses that of $\alpha=1$ after around 700 communication rounds, it lags behind during the earlier phase (below approximately 300 rounds). This fundamentally contradicts with the claim “public dataset sizes have a significant impact on the precision” in line 455.
> > > >
> > > > >In Fig. 5, it appears that $\alpha=0.5$ yields lower accuracy than $\alpha=1$ after 600 rounds. Why would a larger $\alpha$ lead to better accuracy in this case? Why is the accuracy after convergence not monotonically dependent on $\alpha$ in Fig. 5?
> > > >
> > > > >In Table 5, only $\alpha=0$ and $\alpha=1$ are reported regarding time costs. To better understand the computational overhead, it would be helpful to see how it scales across a broader range of $\alpha$ values. Moreover, the statement “would incur prohibitive overhead” lacks empirical support.
> > > >
> > > > >The response does not provide experimental results on how the reconstruction performance changes with $\alpha$, as this cannot be inferred from the current Fig. 5 and Table 5.
> > > >
> > > > We thank the reviewer for the insightful comments. In response, we have significantly expanded our analysis of the $\alpha$ parameter and added new experiments that jointly examine efficiency, model utility, and privacy leakage. The updated Sec. I of the Appendix now includes the following results:
> > > >
> > > > ### Scalability of computational overhead.
> > > > We report new runtime measurements for $\alpha \in$ {$1, 2, 3$} under identical hardware, model, and batch configurations. As shown in Table 8, the training cost increases approximately linearly with $\alpha$, as larger public subsets induce proportionally longer forward–backward passes during each local update. These measurements provide direct empirical support for our earlier claim that large $\alpha$ values introduce increasing computational overhead.
> > > >
> > > > ### Extended accuracy trajectories across a broader range of $\alpha$.
> > > > Figure 8 now presents accuracy curves for $\alpha \in$ {$0.01, 0.1, 0.5, 1, 2, 3$}. We observe a consistent trend: larger $\alpha$ injects more noise in the early training stage, leading to less stable accuracy trajectories. As training progresses, however, these curves gradually stabilize, and the final accuracy differences become small.
> > > > We further clarify that the non-monotonicity is expected in our FL setting because only 10% of clients participate in each communication round. This stochastic participation introduces optimization variance that can obscure a clean monotonic relationship between $\alpha$ and the converged accuracy.
> > > >
> > > > ### Privacy leakage as a function of $\alpha$.
> > > > We additionally evaluate reconstruction performance under GI-NAS for $\alpha \in$ {$1, 0.75, 0.5, 0.25$} (Table 9). The reconstruction metrics differ only slightly across these settings, indicating that privacy benefits saturate once the public subset reaches a sufficiently large size—even though runtime continues to scale with $\alpha$.
> > > >
> > > > ## Response to Q3 and Q4
> > > > >Where are the experiments corresponding to the statement: “In particular, we use FFHQ as the public dataset for CIFAR-10, where the two datasets have completely disjoint semantics and visual characteristics”? In the current submission (Fig. 2 and Table 2), I only see three private/public dataset pairs (CIFAR-10/ImageNet, ImageNet/CIFAR-10, FFHQ/UTKFace), which are fundamentally different from what the authors claim.
> > > >
> > > > >It is recommended to include experiments where the semantic gap between the private and public datasets is further increased.
> > > >
> > > > Thank you for pointing this out. To address your concern more explicitly, we have added a dedicated set of experiments in Sec. J of the Appendix. These experiments use CIFAR-10 paired with FFHQ as the public dataset, creating a substantially larger semantic and visual gap than the pairs reported in the main text.
> > > >
> > > > These additional results demonstrate that our method remains stable even under this extreme mismatch scenario, and the observed trends are consistent with our main paper's conclusions. While this pairing is not a standard evaluation setting in the literature, we agree that it helps clarify the robustness of our approach under large domain shifts, and have therefore included the full experimental details and analysis in Sec. J of the Appendix

---

> > > > > ### Author Response · Authors · 2025-11-30
> > > > >
> > > > > ## Response to Q3 and Q4
> > > > > >Where are the experiments corresponding to the statement: “In particular, we use FFHQ as the public dataset for CIFAR-10, where the two datasets have completely disjoint semantics and visual characteristics”? In the current submission (Fig. 2 and Table 2), I only see three private/public dataset pairs (CIFAR-10/ImageNet, ImageNet/CIFAR-10, FFHQ/UTKFace), which are fundamentally different from what the authors claim.
> > > > >
> > > > > >It is recommended to include experiments where the semantic gap between the private and public datasets is further increased.
> > > > >
> > > > > Thank you for pointing this out. To address your concern more explicitly, we have added a dedicated set of experiments in Sec. J of the Appendix. These experiments use CIFAR-10 paired with FFHQ as the public dataset, creating a substantially larger semantic and visual gap than the pairs reported in the main text.
> > > > >
> > > > > These additional results demonstrate that our method remains stable even under this extreme mismatch scenario, and the observed trends are consistent with our main paper's conclusions. While this pairing is not a standard evaluation setting in the literature, we agree that it helps clarify the robustness of our approach under large domain shifts, and have therefore included the full experimental details and analysis in Sec. J of the Appendix
> > > > >
> > > > > ## Response to W5
> > > > > >Why is there no table or figure for the ablation studies comparing (1) and (2)? The authors only provided a textual description without showing any experimental results. Additionally, the ablation experiment on using `the highest similarity` (instead of taking the average) was not provided.
> > > > >
> > > > > Thank you for the helpful suggestions. We would like to clarify that our use of the 80 prompt templates follows the default CLIP setup and is not intended as a new design choice requiring an ablation study. Template ensembling is a standard practice in CLIP, known to provide more stable zero-shot representations across classes.
> > > > >
> > > > > Our original wording (“We compare …”) may have inadvertently suggested that we were introducing a novel variant to evaluate. In fact, the 80-template setting is simply the CLIP default, and we only aimed to explain the rationale for adopting it. To avoid any ambiguity, the revised manuscript now explicitly states this.
> > > > >
> > > > > Separately, we have conducted ablation experiments on the label alignment strategy itself. Table 4 reports the impact of different alignment strategies on reconstruction performance, and the corresponding discussion is provided in Sec. 3.4.

---

> > > > > > ### Author Response · Authors · 2025-12-01
> > > > > >
> > > > > > ##  Response to W3
> > > > > > >In line 237, the authors claim that $\hat{\mathcal{H}}_{i}$ is independent of $\mathcal{D}$. However, the distributional gap between the public and private datasets does not seem to be significant in the experiments of this paper (i.e., CIFAR-10 vs. ImageNet, FFHQ vs. UTKFace). For instance, both FFHQ and UTKFace are facial datasets and share some distributional similarities. When the distribution of the public dataset is given, the uncertainty regarding the distribution of the private dataset decreases. So they cannot be simply identified as “independent”. With this problematic assumption, the theoretical results of this paper become questionable.
> > > > > >
> > > > > > We sincerely thank the reviewer for pointing out the inaccuracy in our statement: *“Intuitively, since the perturbation $\xi_{\mathcal{M}}$ is sampled from public data $\hat{\mathcal{H}}{i}$ that is independent of $\mathcal{D}{i}$, injecting it cannot increase the mutual information.”*
> > > > > >
> > > > > > We agree that this phrasing is problematic, as the public and private datasets could share distributional similarities (e.g., both being facial datasets) and should not be considered simply "independent."
> > > > > >
> > > > > > As suggested, we remove this specific sentence in the revised manuscript to avoid any confusion or misunderstanding.
> > > > > >
> > > > > > We emphasize that removing this intuitive explanation does not affect the validity or completeness of our theoretical framework. Our formal analysis and proofs rely on $\xi_{\mathcal{M}} \bot\bot \mathcal{D}{i} \mid (\theta^{\tau}, \theta_{\mathcal{D}_{i}}^{\tau})$, rather than requiring the underlying data distributions to be statistically independent.
> > > > > >
> > > > > > We appreciate this sharp observation, which has helped us improve the rigor and precision of our paper.

---

### Official Review · Reviewer_uZ4D · 2025-10-31

**Soundness:** 2
**Presentation:** 3
**Contribution:** 2
**Rating:** 4
**Confidence:** 4

**Summary:**

Recognizing that gradients in federated learning are highly vulnerable to inversion attacks and that standard defenses inevitably degrade model utility, this paper proposes GradientHide, which introduces a two-stage local update strategy that utilizes public data to obscure the gradients.

**Strengths:**

1. This paper combines the idea of mixing up public data and private data to counter the gradient inversion attacks. Utilizing CLIP for label alignment is new.

**Weaknesses:**

1. The experiments are conducted using only ResNet-18, and it is not clear if the proposed method works for other networks. Moreover, for the evaluation of privacy protection, it is not clear if multiple local steps are enabled.
2. The comparison with existing methods does not seem fair from two perspectives. First, the proposed method essentially requires additional SGD steps, which means that measuring the model performance in terms of communication rounds is not fair. In addition, adding new training data may also improve the test accuracy. Second, additional local updates naturally make it more difficult for gradient inversion attacks. It is not clear if the improved privacy is due to the proposed method or simply the additional SGD step.
3. In Tables 2 and 3, there are some minor mistakes in the results of LPIPS. The results highlighted in bold are not necessarily the best.

**Questions:**

1. How many local steps are conducted in the evaluation of privacy protection?
2. Is the comparison fair? Could the authors compare with existing methods that incorporate additional public data?

---

> ### Author Response · Authors · 2025-11-19
>
> ### Response to Weakness \#1 and Question \#1:
>
> In Sec. 3.2 Privacy Protection, we set the number of local steps to 1. This is the most standard and strictest setting used in the GIA literature to evaluate leakage from a single gradient. We will explicitly state this setup in Sec. 2.1 Threat Model of the revised manuscript.
>
> ### Response to Weakness \#1 and Question \#2:
>
> Our choice of ResNet-18 was intentional. Prior work and existing SOTA methods in gradient inversion methods commonly use ResNet-18, so we followed this convention for a fair comparison. Consequently, we also evaluated model performance using the same architecture. To ensure consistency, we adopted the experimental configurations of the corresponding baselines: CENSOR for privacy protection (Sec. 3.2) and FedSAM [1] for model performance (Sec. 3.3).
>
> More critically, our method is architecture-agnostic by design. Our technique operates at the gradient-level, not the model-level. It protects privacy by obfuscating the client's private gradient with a public gradient. This mechanism is independent of the specific network architecture (e.g., ResNet that produced those gradients).
>
> ### Response to Weakness \#3:
>
> Thank you for pointing out these errors in the presentation of results. We will correct them in the revised version.
>
> ### References:
>
> [1] Qu, Zhe, et al. "Generalized federated learning via sharpness aware minimization." International conference on machine learning. PMLR, 2022.

---

### Official Review · Reviewer_TL9A · 2025-11-01

**Soundness:** 2
**Presentation:** 2
**Contribution:** 2
**Rating:** 4
**Confidence:** 3

**Summary:**

The paper tackles the privacy vulnerability in federated learning (FL) where shared gradient updates can be exploited by gradient inversion attacks (GIAs) to reconstruct clients’ private data. To address this, the authors propose GradientHide, a defense framework that introduces a two-stage local update: after training on private data, each client performs an additional update using public data to obfuscate sensitive gradient information before transmission. The public data is semantically aligned with private data through CLIP’s zero-shot label alignment, ensuring effective masking without harming model utility. Theoretical analysis shows that this procedure reduces mutual information between gradients and private data, and experiments on CIFAR-10, ImageNet, and FFHQ demonstrate that GradientHide significantly weakens inversion attacks—yielding blurred or semantically shifted reconstructions—while maintaining strong model accuracy even under heterogeneous data distributions.

**Strengths:**

* The work is based on a clearly defined threat model, where the server is honest but curious to explore the client's gradients.
* The proposed method is well motivated by information-theoretic analysis.
* The evaluation on data protection is comprehensive, using 3 metrics and 3 deatasets.

**Weaknesses:**

* The major motivation for the work is that prior arts largely trade model performance for privacy defense (Line 53). However, the claim is problematic. According to Fig 3, most of prior methods have even better performance than the proposed methods when data distribution is not too heterogeneous (beta < 0.5). Even when beta >0.5 (more heterogeous), the peformance degradation is fair compared to the proposed method.
  - Note that ANP has the best protection effects against GIAS, GIFD, SPEAR, and 2nd best on GI-NAS on CIFAR-10 (see Table 1 LPIPS). ANP has almost the same performance in Fig 3. Thus, it is not fair to justify that the proposed method is better than ANP.
* Some of the best values of ANP in Table 1 LPIPS are not highlighted as bold. This essentially misleads readers into thinking ANP is not as good as the proposed method.
* As achieving benign performance is a major claim of the work, the evaluation is quite insufficient. Only one dataset, CIFAR10, was evaluated.
* Not clear if the FL and the attack follows the same learning setup. If not, is it meaningful to discuss the trade-off between model performance and privacy protection?

**Questions:**

* How good is the benign performance of the proposed method on ImageNet and FFHQ?
* Why not compare to InstaHide, which shares similar principle as the proposed method?
* What is the batch size in Section 3.3? Do Sections 3.3 and 3.1.1 follow the same federated learning setup? Specifically, in FL training, are models trained with a batch size of 1?

---

> ### Author Response · Authors · 2025-11-19
>
> ### Response to Weakness \#1:
>
> Thank you for pointing this out. Our statement in Line 53 refers to a general trend reported in prior work: many gradient-perturbation–based defenses reduce model utility due to the injected noise or distortion of gradient signals, as described in [1, 2]. We therefore use the phrasing “often degrade” to characterize this common observation, rather than to imply that all prior GIDs uniformly lead to performance drops.
>
> We agree with the reviewer’s observation that ANP is a strong baseline and maintains high accuracy even when the data distribution exhibits high heterogeneity ($\beta < 0.5$), which aligns with our results in Fig. 3. However, as shown in Table 1, ANP is weaker than our method in terms of PSNR and reconstruction quality under most attacks (GGL, GIAS, GIFD, SPEAR). Since PSNR reflects reconstruction fidelity, this indicates that ANP offers weaker pixel-level privacy protection compared to GradientHide. In contrast, our method provides much stronger privacy defense while maintaining competitive model performance.
>
> ### Response to Weakness \#2:
>
> Thank you for pointing out these errors in the presentation of results. We will correct them in the revised version.
>
> ### Response to Weakness \#3 and Question \#1:
>
> We agree that evaluation on multiple domains strengthens our claims. While we initially focused on CIFAR-10 to follow standard FL protocols [3, 4], we have now expanded our evaluation to include the FFHQ face dataset. We have added these new results in Sec. H of the Appendix. The results confirm that GradientHide maintains high model performance on face data, demonstrating its robustness across different domains.
>
> ### Response to Weakness \#4 and Question \#3:
>
> Thank you for raising this critical question. Yes, these two setups (Sec 3.2 Privacy Protection and Sec 3.3 Model Performance) are intentionally different. We argue that this distinction is crucial for a rigorous evaluation of the trade-off: privacy protection must be demonstrated under a worst-case scenario, while model performance should be evaluated under a realistic scenario.
>
> * Sec 3.2 (Privacy Protection): To evaluate the worst-case privacy leakage, we adopted the extreme setting most favorable to the attacker (details in Sec. 3.1.1). This is the standard practice in the Gradient Inversion Attack (GIA) literature (details in Sec. 2.1 Threat Model), as it maximizes the single-sample information leaked in the gradient.
>
> * Sec 3.3 (Model Performance): To evaluate model utility in a practical FL setting, we followed the standard FL configuration in [3, 4]. The 50,000 training samples in CIFAR-10 are partitioned across 100 clients under the specified data-heterogeneity setting, giving each client 500 samples. Accordingly, we set the per-client batch size to 500, which aligns with the typical scale of local datasets used in CIFAR-10 FL benchmarks and avoids introducing additional tuning variables.
>
> ### Response to Question \#2:
>
> Thank you for this suggestion. While both methods leverage public data, they are fundamentally different. InstaHide [5] is an input-level encoding technique, whereas our method is a gradient-level defense.
>
> The primary reason we do not compare them is that InstaHide has been proven vulnerable to adaptive attacks. As shown by Carlini et al. [6], InstaHide can be fully broken. Under a white-box assumption (like Sec. J in the Appendix), the server is the attacker and is assumed to know the public dataset (as it is responsible for distributing it). Therefore, InstaHide is provably ineffective under the threat model considered here. Thus, we focus on other GIDs.
>
> ### References:
>
> [1] Martin Abadi, Andy Chu, Ian Goodfellow, H Brendan McMahan, Ilya Mironov, Kunal Talwar, and Li Zhang. Deep learning with differential privacy. In Proceedings of the 2016 ACM SIGSAC Conference on Computer and Communications Security, pp. 308–318, 2016.
>
> [2] Kang Wei, Jun Li, Ming Ding, Chuan Ma, Hang Su, Bo Zhang, and H Vincent Poor. User-level privacy-preserving federated learning: Analysis and performance optimization. IEEE Transactions on Mobile Computing, 21(9):3388–3401, 2021.
>
> [3] Qu, Zhe, et al. "Generalized federated learning via sharpness aware minimization." International conference on machine learning. PMLR, 2022.
>
> [4] Ziqing Fan, Shengchao Hu, Jiangchao Yao, Gang Niu, Ya Zhang, Masashi Sugiyama, and Yanfeng Wang. Locally estimated global perturbations are better than local perturbations for federated sharpness-aware minimization. In International Conference on Machine Learning, pp. 12858–12881. PMLR, 2024.
>
> [5] Huang, Yangsibo, et al. "Instahide: Instance-hiding schemes for private distributed learning." International conference on machine learning. PMLR, 2020.
>
> [6] N. Carlini et al., "Is Private Learning Possible with Instance Encoding?," 2021 IEEE Symposium on Security and Privacy (SP), San Francisco, CA, USA, 2021, pp. 410-427, doi: 10.1109/SP40001.2021.00099.

---

> > ### Comment · Reviewer_TL9A · 2025-11-23
> > **Discussion**
> >
> > Thank you for your rebuttal, which addressed some of my concerns.
> >
> > > privacy protection must be demonstrated under a worst-case scenario
> >
> > I agree that the worst-case attacking evaluation is reasonable for research.
> > However, will such a "worst case" actually happen? Is there any way to simply exclude such a scenario by rules (limiting the local iterations, etc.)? If it is not possible, then it is valuable to examine the model performance when such a scenario could ever happen (even it only happened on a few clients).
> >
> > > Thank you for this suggestion. While both methods leverage public data, they are fundamentally different. InstaHide [5] is an input-level encoding technique, whereas our method is a gradient-level defense.
> > >
> > > The primary reason we do not compare them is that InstaHide has been proven vulnerable to adaptive attacks. As shown by Carlini et al. [6], InstaHide can be fully broken. Under a white-box assumption (like Sec. J in the Appendix), the server is the attacker and is assumed to know the public dataset (as it is responsible for distributing it). Therefore, InstaHide is provably ineffective under the threat model considered here. Thus, we focus on other GIDs.
> >
> > Considering the similar principle between GradientHide and InstaHide, it is necessary to elaborate why gradient-level defense is robust against attacks like [6]. One major lession from [6] is that Instance Encoding is impossible. Note that GradientHide is also instance encoding. For readers who are not familiar with [6], it will be confusing to claim that GradientHide is safer while it shares similar instance encoding principles.

---

> ### Author Response · Authors · 2025-11-24
>
> Thank you for your comments!
>
> > I agree that the worst-case attacking evaluation is reasonable for research. However, will such a "worst case" actually happen? Is there any way to simply exclude such a scenario by rules (limiting the local iterations, etc.)? If it is not possible, then it is valuable to examine the model performance when such a scenario could ever happen (even it only happened on a few clients).
>
> ### Will such a "worst case" actually happen?
> While the exact worst-case scenario may not occur frequently in practice, it is not unrealistic—especially in real-world FL deployments, where many clients inherently have very small local datasets. In practical FL applications (e.g., hospitals, mobile devices, small institutions), some clients naturally possess only a few hundred samples or even fewer. These small-data clients have small local batch sizes, which directly places them at higher risk of gradient inversion.
>
> Furthermore, recent advances in gradient inversion attacks show that attackers no longer rely solely on extreme edge cases. Even medium-to-large batch sizes can be inverted without any defenses. For example:
>
> * SPEAR [7] can exactly reconstruct images even with a batch size of 100, achieving a PSNR of 81 dB on Tiny ImageNet.
> * GI-NAS [8] achieves a PSNR of 31 dB on CIFAR-10 with a batch size of 96.
>
> These results demonstrate that the attack surface extends well beyond the “worst case”, and small-batch clients in real FL systems remain genuinely vulnerable. Thus, evaluating privacy protection in the worst-case scenario is necessary to ensure that even the most disadvantaged clients are protected, which is central to FL’s security guarantees.
>
> ### Is it possible to exclude such a scenario by rules (e.g., limiting local iterations)?
> In our threat model, the server itself is the honest-but-curious attacker. Because the server receives all client updates and controls the aggregation process, it can always choose configurations that maximize its ability to perform inversion. As a result, constraints enforced by the server—such as mandating larger batch sizes or restricting local iterations—cannot be relied upon as a defense, since the attacker can simply ignore or bypass them.
>
> If we step outside this threat model and assume a non-adversarial server, then using larger batch sizes would indeed mitigate inversion risk and generally improve global model performance. This is consistent with standard FL practice, and for this reason, our model performance evaluations focus on the canonical FL training setup, including the class-heterogeneous scenario, which reflects how different GIDs behave in realistic FL conditions.
>
> [7] Dimitar I Dimitrov, Maximilian Baader, Mark M ̈uller, and Martin Vechev. Spear: Exact gradient inversion of batches in federated learning. Advances in Neural Information Processing Systems, 37:106768–106799, 2024.
>
> [8] Wenbo Yu, Hao Fang, Bin Chen, Xiaohang Sui, Chuan Chen, Hao Wu, Shu-Tao Xia, and Ke Xu. Gi-nas: Boosting gradient inversion attacks through adaptive neural architecture search. IEEE Transactions on Information Forensics and Security, 2025.

---

> ### Author Response · Authors · 2025-11-24
>
> Thank you for your comments!
>
> > Considering the similar principle between GradientHide and InstaHide, it is necessary to elaborate why gradient-level defense is robust against attacks like [6]. One major lession from [6] is that Instance Encoding is impossible. Note that GradientHide is also instance encoding. For readers who are not familiar with [6], it will be confusing to claim that GradientHide is safer while it shares similar instance encoding principles.
>
> ### Why can Carlini et al. [6] break InstaHide?
> The attack in [6] relies on the fact that InstaHide exposes input-level instance encodings, where each encoded image is a nearly linear mixture of a private image and several public images. Since the attacker observes these encoded inputs and knows the public dataset, [6] can cluster the encoded images in the feature space, recover the corresponding public components, and then invert the linear mixing to reconstruct the private image. Thus, the result in [6] specifically targets defenses that leak encoded instances.
>
> ### Why does [6] not apply to GradientHide?
> GradientHide does not expose any encoded inputs or intermediate representations. The server only receives a final gradient, which is a highly non-linear aggregation of two local optimization stages. The gradient is not a nearly linear mixture of private and public samples, and it does not preserve instance-level linear structure required by [6]. As a result, the core assumption enabling [6]—access to encoded instances whose components can be matched against the public dataset—does not hold.
>
> Therefore, even though both methods use public data, GradientHide does not perform instance encoding in the sense defined by [6]. The result of [6] concerns input-level instance encoding, while GradientHide operates purely in gradient space, where such linear recoverability does not exist. For this reason, attacks like [6] are inapplicable to our method.

---

> > ### Comment · Reviewer_TL9A · 2025-11-25
> > **Discussion**
> >
> > Thank you for the responses. My concerns are clear. Please update the paper to clarify the points.
> > I have updated my rate.

---

> > > ### Author Response · Authors · 2025-11-25
> > >
> > > Dear Reviewer TL9A,
> > > Thanks for your valuable comments, and we are pleased to know that our responses have clarified your concern.
> > >
> > > We will certainly revise the paper to clarify the raised points.
> > >
> > > Sincerely,
> > > The authors

---

> > > > ### Author Response · Authors · 2025-11-25
> > > >
> > > > We have revised the paper to include a detailed discussion about InstaHide in Sec. A.2 of the Appendix.

---

### Meta-Review · Area_Chair_noUw · 2026-01-11

**Summary:**

Across reviewers, the paper was generally regarded as solid but borderline. All reviewers initially assigned scores just below the acceptance threshold, while indicating openness to acceptance if the identified concerns were adequately addressed. However, several recurring issues ultimately led to the decision to reject. The main concerns were about the privacy–utility trade-off, the degree of novelty beyond careful engineering and extensive evaluation, and the overall rigor and clarity of the presentation. Reviewers questioned whether the contribution is sufficiently distinct and convincingly demonstrated to meet the bar of a top-tier venue. In particular, multiple reviewers noted that the core idea—leveraging public data for gradient obfuscation—appears to constitute careful engineering rather than a fundamental technical advance, and that the contribution seems incremental relative to prior work such as InstaHide. In addition, a significant methodological concern was raised regarding the fairness of the experimental comparisons. The proposed method introduces additional SGD steps and access to extra training data, which may confer an inherent advantage and make direct comparisons with baselines potentially misleading without stronger controls or normalization.

**Reviewer Concerns:**

The rebuttal clarified the threat model and evaluation protocol (worst-case privacy vs. realistic utility), justified robustness relative to InstaHide, corrected presentation errors, added computational cost comparisons, expanded sensitivity analysis for key parameters, and removed an overreaching theoretical independence claim—resolving most technical and clarity issues.

**Reviewer Scores:**

While some of concerns resolved during rebuttal, the reviewers sill believed novelty remains incremental, the threat-model is not clear enough, and the paper requires extensive empirical validation.

---

### Decision · Program_Chairs · 2026-01-26

Reject